# Metabolic and inflammatory biomarker trajectories after a cancer diagnosis and the risk of cardiovascular diseases

Hyemi Park[1], Quan Wang[2], Qianwei Liu [3], Jing Wu[1], Kelu Yang[4], Xinyu Zhang[5], Saro H. Armenian[6,7], Wenjiang Deng[8], Niklas Hammar[1], Maria Feychting [1], Fang Fang [1], Kejia Hu [1,9,10] ✉ & Dang Wei [1,11]

Cancer patients face a high comorbidity burden of cardiovascular diseases (CVD). There is little evidence about stratifying cancer patients for their CVD risk. Here, we conduct a cohort study (1985-2020) to examine post-cancer biomarker trajectories and their associations with subsequent CVD risk. We identify biomarker trajectories after cancer diagnosis for 11 biomarkers using latent class growth modelling. Through Cox regression, we find that cancer patients with initially high and subsequently increasing glucose levels have a 110% higher CVD risk than those with low-stable levels. A low-stable albumin level, even within the normal range, is associated with elevated CVD risk, whereas patients with initially low-stable but subsequently increasing uric acid levels exhibit a 20% reduced risk of CVD. These findings suggest that specific trajectories of glucose, albumin, and uric acid are associated with CVD risk. Further research should elucidate underlying biological mechanisms and validate these associations to improve CVD prevention in cancer patients.

Advancements in early detection and cancer treatment have significantly increased the survival rates for many cancer patients[1,2]. Consequently, increased attention is given to common comorbidities of cancer, such as hospital treated cardiovascular diseases (CVD)[3–5]. Notably, patients with cancer of a favourable prognosis (e.g., 5-year survival rate ≥80%) are more prone to die from CVD than from cancer itself[6–8].

The identification of individuals at increased risk of CVD is crucial for preventing future cardiovascular events in cancer patients. Although the quality of evidence is insufficient or intermediate, recent clinical guidelines moderately to strongly recommend the use of echocardiography, cardiac imaging, and cardiac serum biomarkers to estimate the risk of CVD during and after cancer treatment[9–11]. However, no recommendations were made regarding the frequency and duration of the surveillance due to a lack of evidence, presenting a significant challenge for the precision prevention of CVD in cancer patients in practice.

The elevated risk of CVD in cancer patients is often attributed to a persistent pro-inflammatory status and vascular toxicity resultant of cancer treatment[11], apart from shared risk factors such as aging, smoking, obesity, and diabetes[12,13]. The treatments can negatively impact the immune, cardiovascular, and endocrine systems, leading to increasing inflammation and abnormalities in lipid, apolipoprotein, and carbohydrate metabolisms[14]. These alterations may become

[1]Institute of Environmental Medicine, Karolinska Institutet, Stockholm, Sweden. [2]Ambulatory Surgery Center, Xijing Hospital, the Fourth Military Medical University, Xi'an, China. [3]Department of Hematology, Nanfang Hospital, Southern Medical University, Guangzhou, China. [4]Department of Public Health and Primary Care, Academic Centre for Nursing and Midwifery, Katholieke Universiteit Leuven, Leuven, Belgium. [5]West China School of Nursing, Sichuan University, Chengdu, China. [6]Department of Population Sciences, City of Hope Comprehensive Cancer Center, Duarte, CA, USA. [7]Department of Pediatrics, City of Hope Comprehensive Cancer Center, Duarte, CA, USA. [8]Department of Environmental Health, Harvard T.H. Chan School of Public Health, Boston, MA, USA. [9]Sleep Medicine Center, Mental Health Center, National Center for Mental Disorders, Sleep Research Laboratory, West China Hospital, Sichuan University, Chengdu, China. [10]West China Biomedical Big Data Center, West China Hospital, Sichuan University, Chengdu, China. [11]Department of Social and Behavioral Sciences, Harvard T.H. Chan School of Public Health, Boston, MA, USA. ✉e-mail: kejia.hu@ki.se

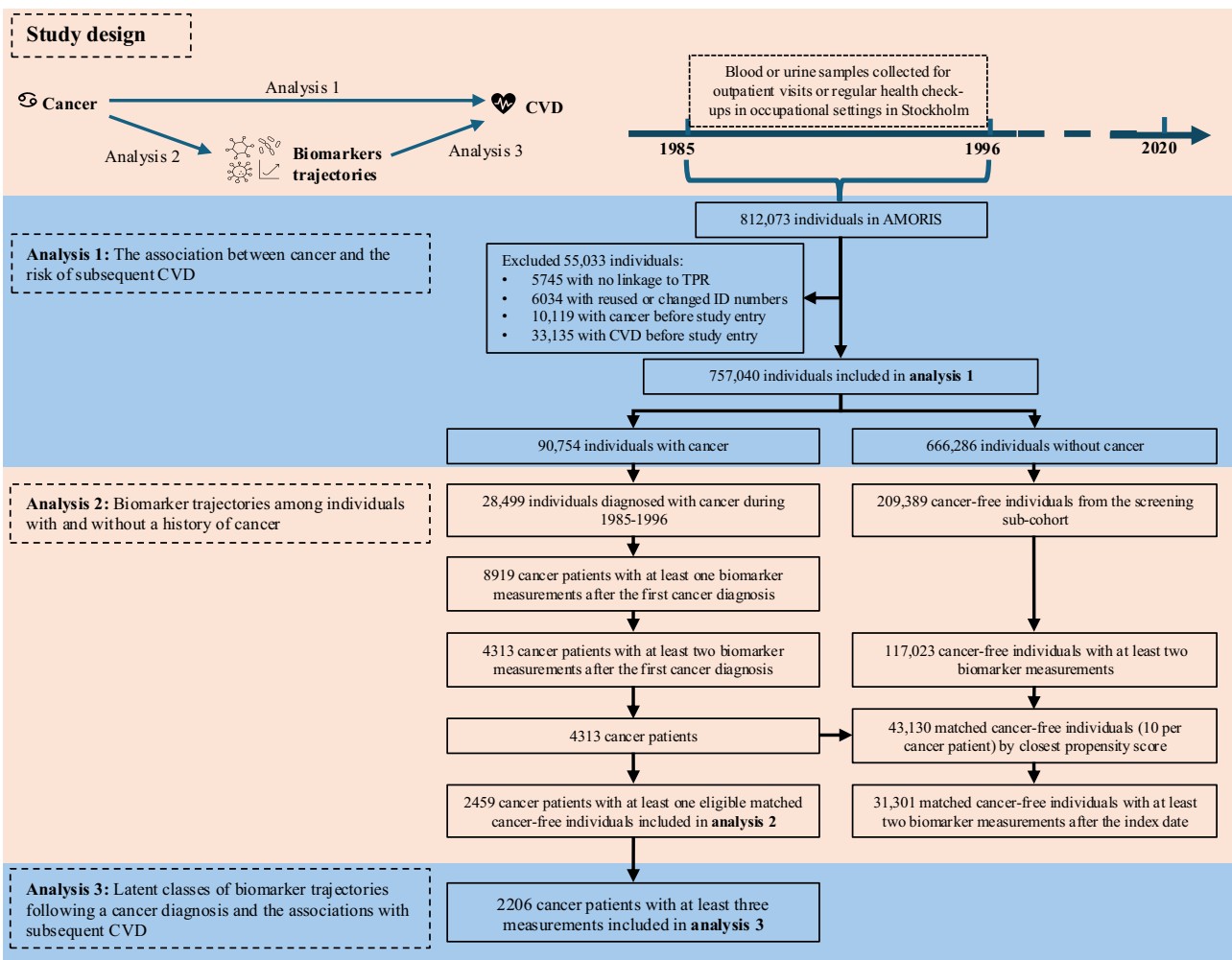

Fig. 1 | **Study design and analytical approaches.** CVD cardiovascular disease, TPR Total Population Register.

chronic and, in turn, increase the risk of subsequent CVD[14]. Previous studies consistently reported higher low-density lipoprotein (LDL) levels and lower high-density lipoprotein (HDL) and apolipoprotein A1 (ApoA1) levels in cancer patients compared to cancer-free individuals during and shortly after treatment[15,16]. Additionally, research on triglycerides (TG) and high-sensitivity C-reactive protein (CRP) also showed different trends between cancer patients and cancer-free individuals although the directions are inconsistent[16–19]. The prior efforts indicate that monitoring metabolic and inflammatory biomarkers through normal blood tests, an easier and less costly approach than those recommended by guidelines, may have a great implication in CVD prevention among cancer patients.

The evidence is, however, very limited on longitudinal changes in metabolic and inflammatory biomarkers after a cancer diagnosis. Previous studies have small sample sizes (ranging from 12 to 729 participants) and varying lengths of follow-up, such as during cancer therapy or maximum 1–5 years after its completion[15–19]. The methodological limitations highlight the need for a large-scale study to gain a comprehensive understanding of metabolic and inflammatory changes following a cancer diagnosis. Additionally, no study has tried to identify latent classes of biomarker trajectories and investigate the associations of such classes with cardiotoxicity or clinical cardiovascular events following a cancer diagnosis.

In this work, we take advantage of the Swedish AMORIS (Apolipoprotein-related MOrtality RISk) cohort to evaluate whether the risk of CVD and longitudinal levels of blood metabolic and inflammatory biomarkers differed across individuals with and without cancer, and whether cancer patients at increased risk of CVD could be identified through repeated measurements of biomarkers in normal blood tests. Specifically, we aim to analyze (1) whether the association between cancer and the increased risk of CVD was evident in this cohort, (2) longitudinal biomarker trajectories among individuals with and without a history of cancer, and (3) latent classes of biomarker trajectories post-cancer diagnosis and their associations with CVD risk. We find that cancer diagnosis is associated with increased risk of CVD, and cancer patients exhibit distinct patterns of glucose, fructosamine, TG, HDL, ApoA1, CRP, haptoglobin, and uric acid levels compared to cancer-free individuals. Specific trajectories of glucose, albumin, and uric acid are associated with CVD risk.

## Results

### Analysis 1: cancer and CVD risk

Among the 757,040 individuals included in this analysis, 90,754 (12.0%) had a cancer diagnosis during follow-up (Fig. 1). Compared to individuals without cancer, cancer patients were more likely to be born in Sweden, older, employed, and have higher income at the first blood sampling (Supplementary Dataset 1, Source Data 1).

Individuals with cancer had a 61% higher risk of CVD than those without [IRR (95% CI): 1.61 (1.59–1.62), Fig. 2]. The risk increase of CVD was highest if the cancer was diagnosed before age 18 [IRR (95% CI): 2.54 (1.97–3.28)], based on all studied cancer types. The results remained similar after excluding individuals diagnosed with their first cancer after 1996 (Supplementary Table 1). The top three prevalent CVDs in cancer patients were arrhythmia (mainly atrial fibrillation),

| Exposure | No. events | Rate* | Multivariable model** IRR (95%CI) |
|---|---|---|---|
| Any cancer | | | |
| No | 333127 | 17.5 | Ref. |
| Yes | 45625 | 73.9 | 1.61 (1.59–1.62) |
| Age at cancer diagnosis | | | |
| <18 years | 60 | 10.3 | 2.54 (1.97–3.28) |
| 18–44 years | 2728 | 25.8 | 1.77 (1.70–1.84) |
| 45–54 years | 6096 | 44.9 | 1.66 (1.62–1.71) |
| 55–64 years | 13231 | 72.2 | 1.63 (1.60–1.66) |
| >64 years | 23510 | 125.8 | 1.55 (1.53–1.58) |

**Fig. 2 | Overall associations between any cancer diagnosis and the risk of cardiovascular disease and by age at cancer diagnosis.** IRR, incidence rate ratio; CI, confidence intervals. *per 1000 person-years. **Adjusted for age and calendar period of follow-up, sex, year of birth, country of birth, income, education, and employment status at the first blood sampling, as well as diabetes and psychiatric disorders during follow-up. Incidence rate ratio together with 95% confidence intervals were calculated through Poisson regression models.

| Subtypes of CVD | No. events | Rate* | Multivariable model** IRR (95%CI) |
|---|---|---|---|
| Myocarditis | 187 | 0.2 | 1.77 (1.51–2.08) |
| Heart failure | 16236 | 14.5 | 1.37 (1.35–1.40) |
| Subarachnoid hemorrhage | 507 | 0.4 | 1.35 (1.23–1.49) |
| Intracerebral hemorrhage | 1626 | 1.4 | 1.33 (1.26–1.40) |
| Arrhythmia | 18813 | 17.9 | 1.26 (1.24–1.28) |
| Atrial fibrillation | 16604 | 15.3 | 1.24 (1.22–1.26) |
| Cardiomyopathy | 1009 | 0.8 | 1.24 (1.15–1.33) |
| Stroke | 9900 | 8.7 | 1.17 (1.14–1.20) |
| Ischemic stroke | 8111 | 7.1 | 1.12 (1.09–1.14) |
| Ischemic heart diseases | 12594 | 12.0 | 1.08 (1.06–1.10) |
| Myocardial infarction | 7131 | 6.3 | 1.05 (1.03–1.08) |

**Fig. 3 | Associations between any cancer diagnosis and the risk of cardiovascular disease by subtypes of cardiovascular disease.** CVD, cardiovascular disease; IRR, incidence rate ratio; CI, confidence intervals. *per 1000 person-years. **Adjusted for age and calendar period of follow-up, sex, year of birth, country of birth, income, education, and employment status at the first blood sampling, as well as diabetes and psychiatric disorders during follow-up. Incidence rate ratio together with 95% confidence intervals were calculated through Poisson regression models.

heart failure, and ischemic heart diseases. Cancer was associated with an increased risk of all subtypes of CVD, with the highest IRR noted for myocarditis [IRR (95% CI): 1.77 (1.51–2.08)], followed by heart failure, and subarachnoid hemorrhage (Fig. 3). In the AMORIS cohort, the three most common cancer types by organ system were those affecting the breast and reproductive system, the digestive system, and the hematopoietic system, while the three most common site-specific cancers were breast cancer, colorectal cancer, and melanoma (Supplementary Table 2). Cancer in any organ system was consistently associated with an elevated risk of CVD, with the highest IRR noted for cancer in the respiratory system [IRR (95% CI): 3.55 (3.40–3.71)], followed by hematological malignancy and cancer in the digestive system (Fig. 4). Positive associations were also found for all sites of cancer (Supplementary Fig. 1).

## Analysis 2: cancer and biomarker trajectories
In this analysis, the top three cancer types by organ system were cancers in the breast and reproductive system, the digestive system, and the urinary system (Supplementary Table 3). The median number of biomarker measurements was slightly higher among cancer patients than among cancer-free individuals (Supplementary Table 4). Compared to cancer-free individuals, cancer patients had higher levels of glucose, fructosamine, TG, ApoB/ApoA1 ratio, CRP, haptoglobin, and uric acid but had lower levels of HDL and ApoA1 throughout the follow-up (Fig. 5; Table 1). The LDL/HDL ratio and leukocyte levels in cancer patients increased after the start of follow-up, surpassing those in cancer-free individuals, but by the end of the follow-up, they returned to similar levels as in cancer-free individuals.

## Analysis 3: post-cancer biomarker trajectories and CVD risk
A total of 2206 individuals were included in Analysis 3, among whom the three most frequent cancer types by organ system were the same as those in Analysis 2 (Supplementary Table 5). Biomarkers were measured for up to 12 years following cancer diagnosis, and the highest proportion of individuals tested (40%–50%) occurred within the first five years (Supplementary Fig. 2). Quadratic latent class growth modeling (LCGM) was selected for TC, LDL/HDL ratio, ApoA1, CRP, and uric acid based on the best-fit criteria, while linear models were applied to the rest of the biomarkers (Supplementary Dataset 2). Two classes were identified for glucose (class 1 versus class 2: stable at low levels versus started at high levels and increased), TC (started low with a U-shape pattern versus stable at high levels), HDL (started at low levels and increased versus started at high levels and decreased), LDL (started at high levels and decreased versus started at low levels and increased), LDL/HDL ratio (slight decrease followed by an increase versus slight increase followed by a decrease), ApoA1 (started at high levels and remained stable versus started low levels with an N-shape pattern), ApoB (started at low levels and decreased versus started at high levels and decreased), ApoB/ApoA1 ratio (started at low levels and decreased versus started at high levels and decreased), CRP (started at low levels, rapidly increased, then decreased versus started at high levels, slightly decreased, then increased), uric acid (started at low levels, remained stable, then increased versus started at high levels and remained relatively stable) (Fig. 6). There were three classes in albumin (Class 1–3: low-, moderate-, and high-stable levels), all within the normal range of the population distribution (35–55 g/L)[20].

| Cancer by organ systems | No. events | Rate* | Multivariable model** | IRR (95%CI) |
|---|---|---|---|---|
| Respiratory system | 2061 | 161.9 | 3.55 (3.40–3.71) | |
| Digestive system | 6170 | 105.3 | 1.98 (1.93–2.03) | |
| Hematological malignancy | 3415 | 78.3 | 1.96 (1.89–2.02) | |
| Buccal cavity and pharynx | 764 | 74.5 | 1.80 (1.67–1.93) | |
| Urinary system | 3033 | 92.3 | 1.72 (1.66–1.78) | |
| Breast and reproductive system | 22350 | 69.3 | 1.47 (1.45–1.49) | |
| Others | 6663 | 57.0 | 1.33 (1.29–1.36) | |

**Fig. 4 | Associations between cancer and the risk cardiovascular disease by the subtypes of cancer.** IRR, incidence rate ratio; CI, confidence intervals. *per 1000 person-years. **Adjusted for age and calendar period of follow-up, sex, year of birth, country of birth, income, education, and employment status at the first blood sampling, as well as diabetes and psychiatric disorders during follow-up. Incidence rate ratio together with 95% confidence intervals were calculated through Poisson regression models.

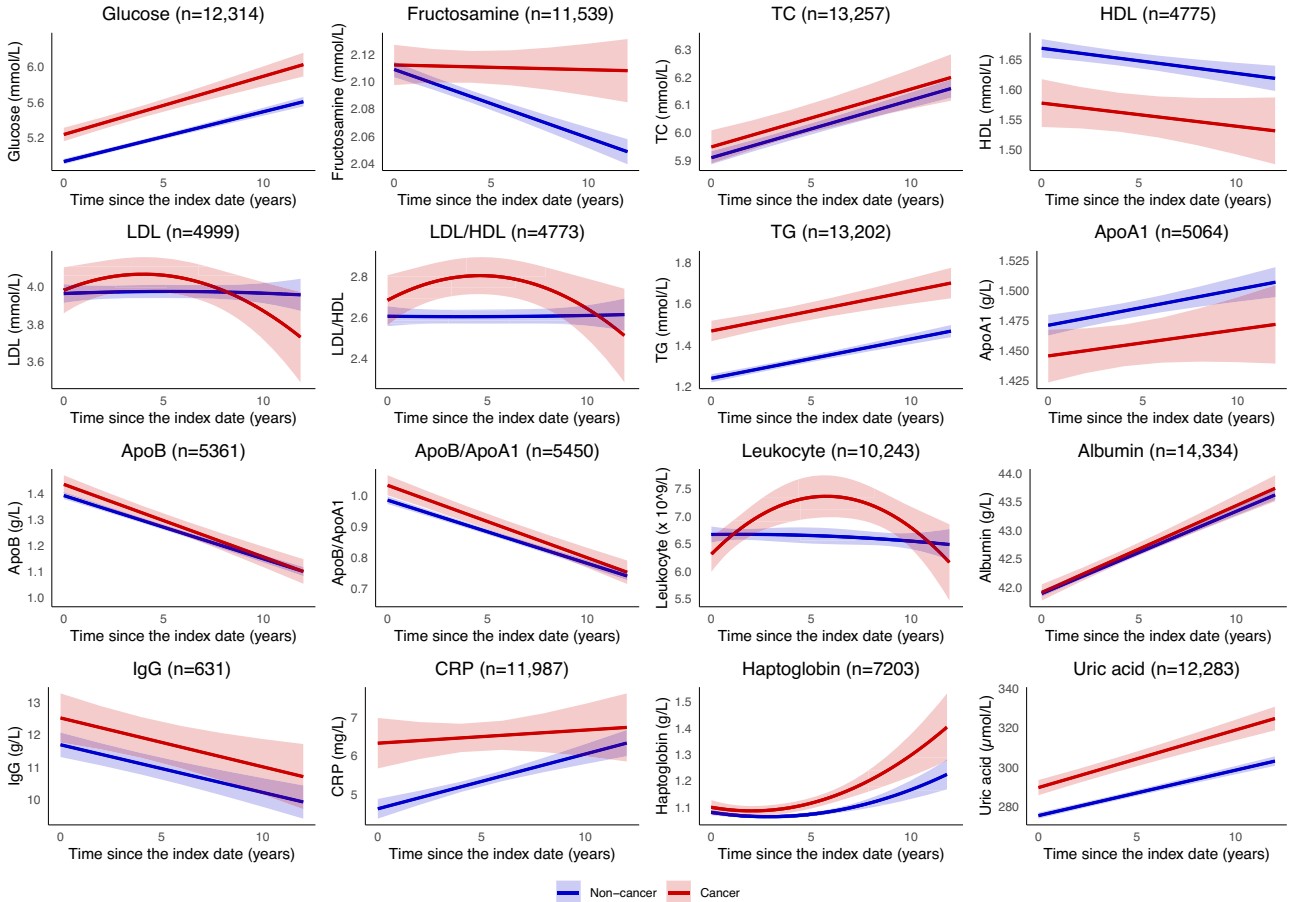

**Fig. 5 | Longitudinal biomarker trajectories among individuals with and without a history of cancer.** TC, total cholesterol; HDL, high-density lipoprotein; LDL, low-density lipoprotein; TG, triglycerides; ApoA1, apolipoprotein A1; ApoB, apolipoprotein B; IgG, immunoglobulin G; CRP, C-reactive protein. Biomarker trajectories in relation to cancer were predicted and plotted by coefficients from the linear mixed effect models with a quadratic term of time for LDL, LDL/HDL ratio, leukocyte, and haptoglobin. Shaded areas represent 95% confidence intervals. Adjusted for propensity score, time since the index date, and interaction between cancer and time.

Compared to cancer patients with stable and low levels of glucose over time, those whose glucose levels started high and increased had a 110% higher risk of CVD [HR (95% CI): 2.10 (1.59–2.78)] (Supplementary Dataset 3). Cancer patients with low-stable levels of albumin had a 56% higher risk of CVD than those with moderate-stable levels [HR (95% CI): 1.56 (1.21–2.01)], although all within the normal range. In contrast, those whose uric acid levels started at low, remained stable and then increased had a 20% lower risk of CVD than cancer patients with high-stable uric acid levels [HR (95% CI): 0.80 (0.70–0.92)]. Regardless of the wide confidence intervals, our results also indicated that cancer patients with specific trajectories in TC and ApoB/ApoA1 ratio were at

an increased risk of CVD. No association, however, was observed for the other biomarkers. Similar results were found for arrhythmia, atrial fibrillation, heart failure, ischemic heart disease, and stroke (Supplementary Dataset 4). Likewise, among cancers of the breast and reproductive system, digestive system, and urinary system (the three most frequent organ system categories in Analysis 3), persistently high glucose levels after cancer diagnosis were associated with an increased CVD risk (Supplementary Table 6). In contrast, among patients with cancer in the breast and reproductive system, persistently low or high albumin levels were associated with approximately 50% lower CVD risk.

**Table 1 | Baseline levels and annual changes in biomarkers among cancer patients compared to cancer-free individuals using linear mixed effects models**

| Biomarker | Coefficients (95% confidence intervals) | | |
|---|---|---|---|
| | Baseline | Annual changes | |
| | | Linear term | Quadratic term |
| Glucose | 0.305 (0.22 - 0.39) | 0.009 (−0.005 - 0.023) | - |
| Fructosamine | 0.003 (−0.013 - 0.02) | 0.005 (0.002 - 0.007) | - |
| TC | 0.039 (−0.028 - 0.106) | 0 (−0.009 - 0.009) | - |
| HDL | −0.091 (−0.136 - −0.047) | 0 (−0.006 - 0.006) | - |
| LDL | 0.054 (−0.041 - 0.149) | −3.597 (−9.643 - 2.45) | −5.172 (−9.916 - −0.428) |
| LDL/HDL ratio | 0.157 (0.062 - 0.251) | −1.332 (−7.114 - 4.45) | −5.892 (-10.4 - −1.384) |
| TG | 0.228 (0.173 - 0.283) | 0 (−0.008 - 0.008) | - |
| ApoA1 | −0.025 (−0.05 - -0.001) | −0.001 (−0.004 - 0.003) | - |
| ApoB | 0.043 (0.005 - 0.082) | −0.004 (−0.009 - 0.002) | - |
| ApoB/ApoA1 ratio | 0.048 (0.012 - 0.083) | −0.003 (−0.007 - 0.001) | - |
| Leukocyte | 0.411 (0.097 - 0.724) | 33.616 (10.134 - 57.099) | −44.881 (−74.416 - −15.347) |
| Albumin | 0.02 (−0.137 - 0.177) | 0.008 (-0.019 - 0.035) | - |
| IgG | 0.83 (−0.08 - 1.739) | −0.004 (−0.116 - 0.108) | - |
| CRP | 1.699 (0.988 − 2.411) | −0.108 (−0.229 - 0.014) | - |
| Haptoglobin | 0.032 (0.012 − 0.052) | 2.622 (0.937 - 4.308) | 1.091 (−0.57 − 2.752) |
| Uric acid | 14.215 (9.879 - 18.551) | 0.615 (0.042 - 1.188) | - |

TC, total cholesterol; HDL, high-density lipoprotein; LDL, low-density lipoprotein cholesterol; TG, triglycerides; ApoA1, apolipoprotein A1; ApoB, Apolipoprotein B; IgG, immunoglobulin G; CRP, C-reactive protein.
Linear mixed effect models with a quadratic term of time were fitted to reflect the changes over time for each biomarker. Adjusted for propensity score, time since the index date, and interaction between cancer and time as fixed effect. Coefficients are presented with 95% confidence intervals derived from linear mixed-effects models.

## Discussion

In a large Swedish cohort with up to 35 years of follow-up, we first confirmed that cancer patients have an increased risk of CVD compared to cancer-free individuals. Cancer patients had higher levels of glucose, fructosamine, TG, CRP, haptoglobin, and uric acid but had lower levels of HDL and ApoA1 within the first 12 years after diagnosis than cancer-free individuals. Among cancer patients, distinct trajectories were observed for 11 metabolic and inflammatory biomarkers. The latent classes of trajectories of glucose, albumin, and uric acid might help identify cancer patients with increased CVD risk.

The present study comprehensively investigates longitudinal metabolic and inflammatory biomarker trajectories and their associations with CVD risk among cancer patients. In line with previous studies[3,5–8,21], we observed a heightened CVD risk after a cancer diagnosis. Our study also showed consistent findings with previous studies, indicating that cancer patients had higher levels of LDL but lower levels of HDL and ApoA1 compared to cancer-free individuals[15,16]. The prior findings regarding longitudinal changes in TG and high-sensitivity CRP are conflicting; some showed declining[16–19], while others showed increasing[17,18] levels. Among over 10,000 individuals analysed for TG and CRP respectively, we observed, in this study, that levels of the two biomarkers were consistently higher in cancer patients than in cancer-free individuals throughout the follow-up. Furthermore, we found that the count of leukocytes was higher in cancer patients than in cancer-free individuals about two years after the diagnosis.

We extend current knowledge by showing that cancer patients also had higher levels of glucose, fructosamine, ApoB/ApoA1 ratio, haptoglobin, and uric acid during the first 12 years after diagnosis, compared to cancer-free individuals. Two previous studies attempted to evaluate the associations between longitudinal changes in several cardiac serum biomarkers such as high-sensitivity cardiac troponin and subsequent cardiotoxicity among breast cancer patients[19,22]. However, they did not focus on latent classes of biomarker trajectories, possibly due to the limited number of biomarker tests and small sample sizes[19,22]. In our study, we identified distinct trajectories of 11

biomarkers among cancer patients, showing that cancer patients at an increased risk of CVD might be identified through investigating longitudinal data on glucose, albumin, and uric acid. The purpose of this study is not to develop clinical guidelines but rather explore if biomarker trajectories can be a promising area for further development of such guidelines. Further investigation, including mechanistic studies and validation in external clinical cohorts are needed before guiding clinical utility. In practice, fine mapping the temporal patterns of these biomarkers may assist in stratifying patients for intensified cardiovascular surveillance and, ultimately, prevention of CVD in the oncology setting.

In line with prior findings[23,24], the present study also found that childhood cancer patients (e.g., diagnosed before 18 years) showed to experience a substantially higher risk of CVD after cancer diagnosis—approximately two- to three-fold greater than that of cancer-free individuals. Given that childhood cancer patients have a relatively longer life expectancy than those patients diagnosed in adulthood, it would be clinically meaningful to examine whether post-cancer biomarker trajectories differ by age at diagnosis and whether their implications vary across age groups. However, we were unable to do so due to small numbers of childhood cancer patients included in this study.

Cancer therapies are known to influence both CVD risk and metabolic biomarkers. For example, cardiotoxicity is a concern for many chemotherapies, especially anthracyclines and certain targeted agents (e.g., Trastuzumab and Tyrosine Kinase Inhibitors)[25]. Chemotherapy, radiotherapy and hormone deprivation therapies have also been suggested to be associated with insulin resistance, dyslipidemia and hyperglycemia[26–29]. Cytotoxic treatments can increase purine turnover through tumor cell lysis, even in the absence of overt tumor lysis syndrome[30]. In addition, renal toxicity, nutritional changes, and systemic inflammation may alter metabolism or excretion of metabolic biomarkers[31,32]. All of these might have contributed to the altered biomarker levels observed in cancer patients compared to non-cancer individuals in the present study, although the biomarker levels

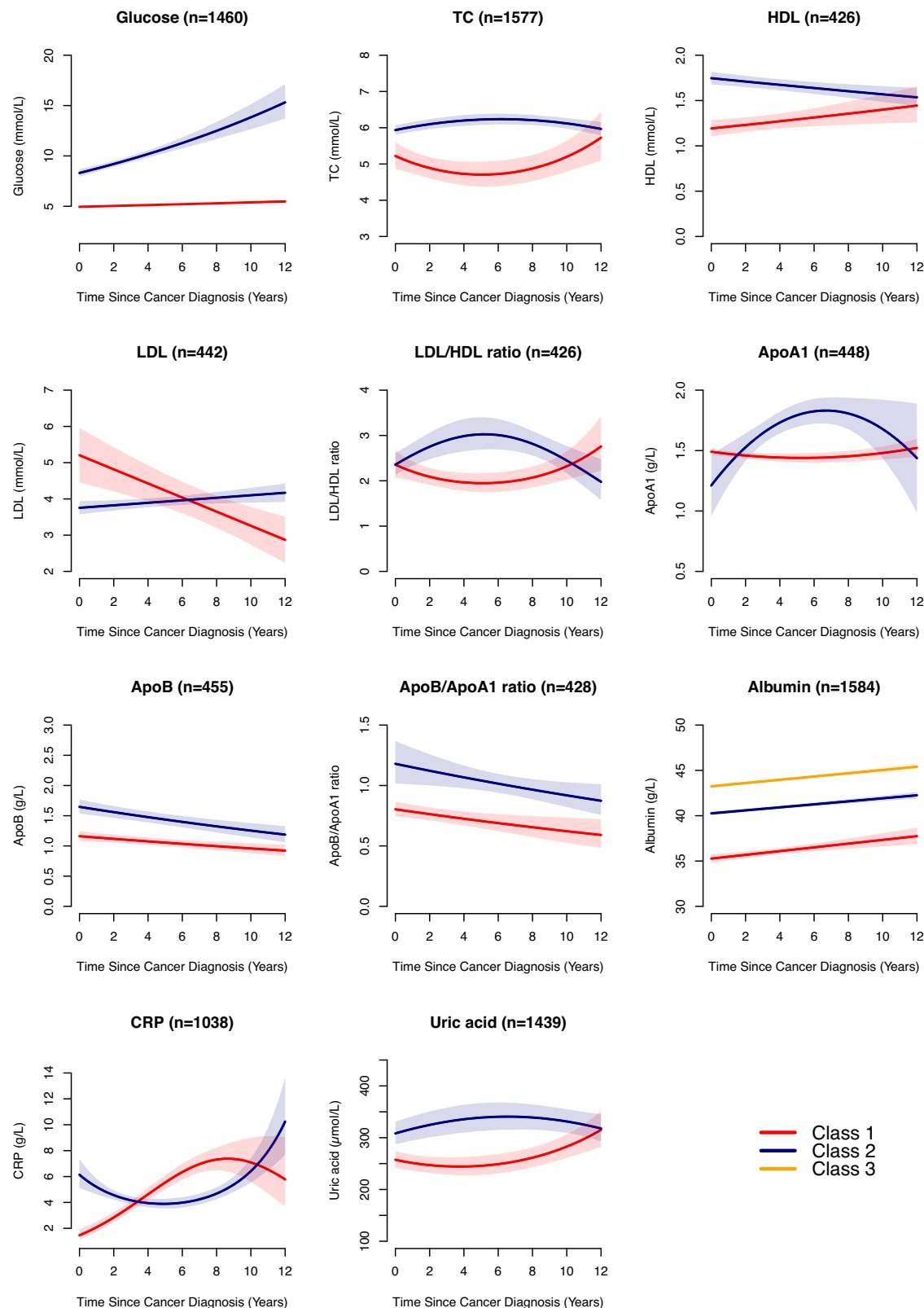

generally remained within the normal ranges. Nonetheless, as our trajectory analyses of biomarkers focused on long-term changes spanning across 12 years after a cancer diagnosis (Supplementary Fig. 2), the potential influence of cancer therapies is limited. Moreover, the observed associations between latent classes of longitudinal trajectories biomarker after diagnosis and CVD risk appear to be largely consistent across the three most frequent cancer types by organ

systems for some biomarkers (e.g., glucose), though not for others (e.g., albumin) (Supplementary Table 6). Further large-scale studies are warranted to confirm the robustness of these associations and to clarify potential sources of heterogeneity.

Altered biomarker levels may represent underlying health conditions and are manageable through corresponding treatments. We adjusted for psychiatric disorders and diabetes in the analyses as they

**Fig. 6 | Latent classes of longitudinal biomarker trajectories following a cancer diagnosis.** Glucose [Class 1: stable at low levels, Class 2: started at high levels and increased]; TC [Class 1: started low with a U-shape pattern, Class 2: stable at high levels]; HDL [Class 1: started at low levels and increased, Class 2: started at high levels and decreased]; LDL [Class 1: started at high levels and decreased, Class 2: started at low levels and increased]; LDL/HDL ratio [Class 1: slight decrease followed by an increase, Class 2: slight increase followed by a decrease]; ApoA1 [Class 1: started at high levels and remained stable, Class 2: started low with an N-shape pattern]; ApoB [Class 1: started at low levels and decreased, Class 2: started at high levels and decreased]; ApoB/ApoA1 ratio [Class 1: started at low levels and decreased, Class 2: started at high levels and decreased]; albumin [Class 1–3: low-, moderate-, and high-stable levels]; CRP [Class 1: started at low levels, rapidly increased, then decreased, Class 2: started at high levels, slightly decreased, then increased]; uric acid [Class 1: started at low levels, remained stable, then increased, Class 2: started at high levels and remained relatively stable]. TC, total cholesterol; HDL, high-density lipoprotein; LDL, low-density lipoprotein; TG, triglycerides; ApoA1, apolipoprotein A1; ApoB, apolipoprotein B; CRP, C-reactive protein. We fitted latent class growth modelling (LCGM) with two polynomial functions of time (linear and quadratic) and three models for biomarker trajectories: fixed effect only, fixed effect with random intercept, and fixed effect with both random intercept and slope. We plotted the optimised trajectories. Shaded areas represent 95% confidence intervals.

were closely related to both the studied biomarkers and risk of CVD. Obesity is also a potential confounder; however, we had limited data on body mass index in the AMORIS cohort. Nevertheless, we speculate that some of these health conditions might be secondary to cancer diagnosis and could thus serve as mediators, rather than confounders, in the association between post-cancer biomarker trajectories and CVD risk, i.e., should not be adjusted for. Treatments for dyslipidemia, dysglycemia and hyperuricemia might, on the other hand, have led to an underestimation of the true association. As the treatments are aiming to modulate the altered biomarker levels to normal, i.e., some cancer patients with abnormal biomarker levels could appear as with "normal" biomarker levels thanks to the treatments. This might have contributed to the null association observed in Analysis 3 for lipids and lipid proteins. Nevertheless, we still found a higher risk of CVD among individuals with persistently high glucose levels. A critical direction for future research is clinical implementation of these biomarker trajectories, taken into consideration cancer therapies, comorbidities and treatment, to tailor cardiovascular prevention strategies in oncology care.

Our study showed that cancer patients with a high start and a continued increase in glucose levels had a higher risk of CVD than those with stable low glucose levels. The trajectory of glucose began with a level higher than the threshold for defining impaired fasting glucose, 5.6–7.0 mmol/L[33], which may suggest an initial metabolic disturbance. Despite being a widely recognised marker of long-term glycemic control, data on HbA1c was available for only 2.3% of the AMORIS participants who have prevalent or suspected diabetes[34] and was thus not included in our analysis. Future studies should incorporate HbA1c, given its strong correlation with fructosamine and glucose in cancer patients despite common oncologic confounders[35]. Fructosamine has been suggested to be a glycemic marker in the AMORIS cohort[36]. Mean of 4 repeated measurements of fructosamine after enrollment to AMORIS, before or after cancer diagnosis, was associated with a reduced cancer risk[37], while a single measurement of fructosamine around 5 months post-diagnosis was linked to an increased risk of death among breast cancer patients[38]. These conflicting findings may suggest that elevated fructosamine levels reflect different underlying metabolic or nutritional states, although both studies have relied on single values of fructosamine. In contrast, we investigated longitudinal patterns of this biomarker following cancer diagnosis and found that, compared to cancer-free individuals, cancer patients showed consistently high levels during the 12 years after cancer diagnosis. However, we did not identify multiple latent classes of fructosamine trajectories among cancer patients, indicating that most patients followed a similar pattern. To the best of our knowledge, AMORIS is one of the largest cohorts worldwide regarding measurements of fructosamine. Taking advantage of the repeated measurements of biomarkers, the current study expands the knowledge for AMORIS cohort by using latent trajectory modeling. The absence of distinct trajectories may be due to our aggregated analysis across all cancer types, which could mask cancer-specific physiological variations.

TC is generally a risk factor for CVD[39,40]. It may also be used to identify cancer patients at high risk of CVD. We found in our study that compared to cancer patients with a stable and high TC level (~6 mmol/L), those with TC levels starting at low, slightly decreasing, and then increasing had an 18% higher risk of CVD though confidence intervals were wide. Similarly, ApoB/ApoA1 ratio and ApoB are found to be associated with the risk of CVD in the general population[41,42], and this relationship may also hold true for cancer patients. Although the ApoB/ApoA1 ratio and ApoB levels in cancer patients were close to that in the cancer-free population in our study, we identified two parallel trajectories (decreased over time) for the two biomarkers respectively among cancer patients. Compared to cancer patients whose ApoB/ApoA1 ratios started at low, those with ApoB/ApoA1 ratios starting at high had a 23% higher risk of CVD although this association is marginal. Similar results were observed in ApoB.

We found that the group of cancer patients with constantly low albumin levels had a 56% higher risk of CVD than those with constantly moderate albumin levels. This finding is in line with existing literature showing that low albumin levels are associated with an increased CVD risk[43,44], suggesting that we could identify cancer patients at high risk of CVD by different levels of albumin even within the normal range of the general population (35–55 g/L)[20]. Consistent with previous findings showing that elevated uric acid levels are linked to an increased risk of CVD[45], we observed that cancer patients whose uric acid levels began low, remained stable, and then increased had a reduced risk of CVD. Several strengths of this study make our findings robust, i.e., large sample size, long follow-up (up to 35 years), repeated biomarker measurements within the long recruitment period (12 years), independent collection and high quality of information on cancer and CVD, a large number of biomarkers, as well as comparisons of CVD risk and biomarker trajectories among cancer patients and cancer-free individuals to support our main findings. These merits enabled us to conduct a comprehensive trajectory analysis and provide valuable insights into the mechanisms linking cancer to CVD.

Our study, however, also has several limitations. First, given that biomarker measurements differed between study participants, both in terms of timing and frequency, there are inherent challenges in accurately modeling these trajectories. However, due to the data-dependent nature of LCGM, such variability might introduce a non-differential misclassification of exposure, which should not systematically favor any trajectories. Second, this study was based on individuals who underwent occupational health check-ups or outpatient laboratory tests in Stockholm, Sweden. Although this does not affect the internal validity of studies derived from the AMORIS cohort, the generalizability of our findings to broader or more diverse populations (e.g., those of non-European ancestry, patients in active treatment settings, or individuals in lower-resource contexts) may be limited. Further research in other populations and settings is warranted to validate our findings. Third, although multiple covariates were adjusted for in the analyses, there may still be residual confounding from unknown or unmeasured factors, e.g., cancer therapies and subclinical cardiometabolic conditions not attended by specialist care, like

atherosclerosis, dyslipidemia, dysglycemia, hyperuricemia, and their treatments. Therefore, caution is warranted when interpreting our findings and considering the clinical utility. Fourth, due to the limit of sample size, we studied cancer as one single category in Analyses 2 and 3, but it is important to keep in mind that cancer is largely heterogeneous in terms of etiology, pathology, treatment and metabolic response. Although we were able to grasp some universal metabolic changes in all cancer types[46], future study should extend the effort to specific cancer types and explore the underlying mechanism in the context of each cancer before clinical implementation. Fifth, attendance to preventive cardiovascular care, including statin use, might be a potential confounder given that cancer survivors may have differential access to it. However, we were unable to find a suitable variable for adjustment. Data on prescribed statin use from the Swedish Prescribed Drug Register have been available only since 2005, which limits the ability to use medication prescription data to identify milder CVD as the outcome of interest. Moreover, statins were not commonly used during 1985–1996, the period in which cancer diagnoses and biomarker measurements were made in our study[47,48]. Therefore, we focused on relatively severe cases of CVD recorded in the Swedish Patient Register. Last, we were unable to perform any stratification analysis by cancer treatment due to the limited availability of treatment data. Future research should expand these efforts to cancer patients receiving different therapies.

In conclusion, cancer patients demonstrated different trajectories of glucose, fructosamine, TG, HDL, ApoA1, CRP, haptoglobin, and uric acid levels over time compared to cancer-free individuals. Cancer patients with specific trajectories of glucose, albumin, and uric acid had an increased risk of CVD. Further research on these blood biomarkers, through demonstrating underlying biological mechanisms and replication in independent cohorts, may contribute to the prevention of cardiovascular disease among cancer patients.

## Methods

This study complies with all relevant ethical regulations and was approved by the Swedish Ethical Review Authority (2018/2401-31, with subsequent amendments). Informed consent is not needed in registry-based research in Sweden.

### Data source and study design

The AMORIS cohort includes 812,073 Swedish residents with laboratory analyses of blood or urine samples collected for outpatient visits or regular health check-ups in the occupational setting in Stockholm during 1985–1996 (i.e., inclusion period)[49]. Study individuals in AMORIS accounted for about 35% of total population living in Stockholm County during the inclusion period and had similar sex distributions but their mean age was four years higher than the general population[49]. The details of the cohort were described elsewhere[49]. Through the unique personal identification number, the AMORIS cohort was linked to several Swedish national registers, including the Patient Register, the Cancer Register, the Causes of Death Register, and the Total Population Register. The Patient Register has collected inpatient care data since 1964 (became nationwide in 1987) and outpatient care data since 2001[50], while the Cancer Register contains information on virtually all cancer diagnoses since 1958.

### Ascertainment of cancer, cardiovascular diseases, and covariates

**Cancer.** Cancer diagnoses were identified from the Swedish Cancer Register using the seventh revision of the International Classification of Diseases (ICD-7) (Supplementary Dataset 5). Despite the introduction of newer ICD versions (ICD-8, ICD-9, and ICD-10) in the Swedish medical record system, all diagnoses in the Cancer Register are recorded using ICD-7 codes. Cancers were categorised into seven organ system groups: buccal cavity and pharynx, digestive system, respiratory system, breast and reproductive system, urinary system, hematological malignancies, and other cancers. Cancers were also categorised by specific sites.

**Cardiovascular diseases.** CVD were identified using primary and secondary diagnoses from inpatient and outpatient records in the Swedish Patient Register, coded using ICD-8, ICD-9, and ICD-10 (Supplementary Dataset 5). We analysed overall CVD and specific subtypes, including ischemic heart disease, myocardial infarction, heart failure, arrhythmia, atrial fibrillation, myocarditis, cardiomyopathy, stroke, ischemic stroke, intracerebral hemorrhage, and subarachnoid hemorrhage.

**Biomarkers.** We focused on 16 commonly measured metabolic and inflammatory biomarkers in the AMORIS cohort, including glucose, fructosamine, total cholesterol (TC), HDL (concentration of HDL-cholesterol), LDL (concentrations of LDL-cholesterol), LDL/HDL ratio, TG, ApoA1, apolipoprotein B (ApoB), ApoB/ApoA1 ratio, albumin, haptoglobin, leukocyte, immunoglobulin G (IgG), CRP, and uric acid.

AMORIS Participants had their first complete profile of ApoB, ApoA-I, TC and TG done simultaneously, and the ApoB/ApoA-I ratio, LDL-cholesterol, and HDL-cholesterol, were then calculated accordingly[51]. Apolipoproteins were measured with immunoturbidimetry. Concentrations of total cholesterol and triglycerides were measured by enzymatic techniques. These methods were fully automated and also specifically described in previous work[52,53]. Concentrations of LDL-cholesterol and HDL-cholesterol were calculated by the Jungner formula: $LDL = 0.48 + 0.99TC - 0.23TG - 1.58ApoA1$; $HDL = TC - 0.45TG - LDL$ for around 85% of the participants. The rest 15% had their LDL calculated by Friedewald formula[54] and HDL measured directly from the blood by an automated precipitation method (Boehringer Mannheim GmbH, Mannheim, Germany)[55].

### Covariates

**Sociodemographic factors.** Data on sex, birth year, and country of birth were obtained from the AMORIS cohort. Data on education, income, and employment were retrieved from the Swedish Population and Housing Census (FOB) of 1985 and the Longitudinal Integration Database for Health Insurance and Labour Market Studies (LISA). LISA includes annual updates from 1990 onward. Educational level, income level, and employment status were defined based on the closest available data before the first blood sampling. Specifically, data from FOB were used for individuals with blood samples collected between 1985 and 1989, and data from LISA were used for those sampled between 1990 and 1996. For individuals with blood samples collected between 1991 and 1996, data from the year before sampling were used if current-year data were missing.

**Comorbidities: psychiatric disorders and diabetes.** Comorbidities of psychiatric disorders and diabetes were identified through primary and secondary diagnoses from in- and outpatient care attended by specialists in the Swedish Patient Register using ICD codes (Supplementary Dataset 5). For diabetes, data on anti-diabetic drug use from the Prescribed Drug Register (available from 2005) were also used.

In the analysis of cancer and CVD risk, we considered psychiatric disorders and diabetes as time-varying variables. Individuals with these diagnoses prior to January 1, 1985, or who never developed them by the end of the study period, were classified as having or not having the conditions throughout the follow-up, respectively. Individuals who developed these conditions during the follow-up period were classified as not having the conditions until diagnosis, and as having them thereafter.

For analyses of biomarker trajectories among individuals with and without a cancer history, a propensity score including psychiatric disorders and diabetes -matched cohort was created for each

biomarker. The history of psychiatric disorders or diabetes was defined using data from the Patient Register up to and including 1985, as cancer diagnoses were recorded between 1985 and 1996.

In the analyses of examining the association between latent classes of biomarker trajectories in cancer patients and subsequent CVD risk, psychiatric disorders and diabetes were defined using specialist diagnoses available up to the end of follow-up. This was because psychiatric disorders and diabetes, even when diagnosed after blood sample collection, may have influenced biomarker levels prior to diagnosis due to their chronic etiological nature and delayed diagnosis.

### Analysis 1: cancer and CVD risk

We conducted a cohort study to evaluate the association between cancer and the risk of CVD including a total of 757,040 individuals [371,286 (49.0%) males, 385,754 (51.0%) females] who were free of cancer and CVD before 1985 in the AMORIS cohort (Fig. 1). We followed the study individuals from January 1st, 1985 until the first CVD diagnosis, emigration, death, or December 31st, 2020, whichever came first. The exposure to cancer was used as a time-varying variable. Individuals who had a cancer diagnosis during the follow-up were followed from study entry until the date of cancer diagnosis as the unexposed group and as the exposed group thereafter. In contrast, those without a cancer diagnosis throughout the follow-up period contributed person-time only to the unexposed groups.

### Analysis 2: biomarker trajectories among individuals with and without cancer

We conducted a matched cohort study per biomarker, including patients diagnosed with the first cancer during 1985–1996 and cancer-free persons individually matched by a propensity score incorporating information on sex, birth year, country of birth, education, income, and employment status at the first blood sampling, as well as psychiatric disorders and diabetes before 1985. To avoid potential indication bias, we selected up to ten cancer-free persons from those who were recruited through health checkups (i.e., the screening sub-cohort) individually per cancer patient using nearest neighbour matching of the propensity score. The date of the cancer diagnosis was defined as the index date for both cancer patients and their individually matched non-cancer individuals. The cancer patients and the non-cancer individuals were required to have at least two measurements of the biomarker after the index date, leading to a total of 2459 cancer patients and 31,301 non-cancer individuals in the analysis, including 13,437 (39.8%) males and 20,323 (60.2%) females.

### Analysis 3: post-cancer biomarker trajectories and CVD risk

We conducted a cohort study per biomarker among cancer patients who had no prior CVD but at least three measurements of the biomarker after cancer diagnosis during 1985–1996. There was in total 2206 individuals [901 (40.8%) males, 1305 (59.2%) females] included in the analysis. Compared to patients with first cancer diagnosis during 1985–1996 and had only one to two measurements of blood biomarkers after diagnosis, no substantial difference in characteristics was observed in those included in this analysis (Supplementary Dataset 6).

### Statistics & reproducibility

**Analysis 1.** We evaluated the incidence rate ratios (IRRs) and 95% confidence intervals (CI) for the CVD risk in relation to cancer using Poisson regression models. We adjusted for age and calendar period at follow-up (split yearly), psychiatric disorders, and diabetes as time-varying variables, as well as sex, birth year, country of birth, education, income, and employment status at the first blood sampling as time-fixed variables in the models. Additionally, we evaluated the association by age at cancer diagnosis, for subtypes of CVD and subtypes of

cancer (organ systems or sites) separately. We also conducted a sensitivity analysis by excluding cancer patients diagnosed after 1996.

**Analysis 2.** We evaluated the trajectories of annual changes of the 16 biomarkers measured during the 12 years (1985–1996) among cancer patients compared to individuals without cancer using linear mixed effect models. We considered cancer, propensity score, time since the index date, the interaction between cancer and time for the fixed effect, as well as time and study participants for the random effect in the models. A quadratic term of time was included to test the non-linear changes in biomarkers. We eventually used a quadratic function of time for LDL, LDL/HDL ratio, leukocyte, and haptoglobin and a linear function of time for the rest biomarkers in the models. Biomarker trajectories in relation to cancer were predicted and plotted by coefficients from the linear mixed effect models.

**Analysis 3.** We first identified latent classes of longitudinal trajectories of the biomarkers using latent class growth modelling (LCGM) and classified cancer patients according to the observed latent classes. We considered two polynomial functions of time (linear and quadratic) and three models for biomarker trajectories: fixed effect only, fixed effect with random intercept, and fixed effect with both random intercept and slope. For each polynomial model, one to five classes were evaluated. We selected the best-fit model for each biomarker which had the lowest BIC with $P < 0.05$ for all polynomial terms and a group membership probability $\geq 5\%$. Given that only 76 cancer patients had three or more measurements of IgG, we did not perform the LCGM analysis for IgG. We did not observe multiple latent classes for fructosamine, TG, leukocyte, and haptoglobin using the criteria, indicating that most cancer patients followed the same trajectory in these biomarkers. Second, we evaluated the associations between trajectories of the remaining 11 biomarkers and the risk of CVD using Cox regression models. The groups with the most cancer patients were defined as the reference. We used attained age as the time scale and adjusted for sex, year at cancer diagnosis, country of birth, education, income, employment status at the first blood sampling, psychiatric disorders, and diabetes by the end of follow-up in Cox models. The analyses were performed for any CVD and the subtypes. Due to small numbers, cancer types were broken down by system in Analysis 2 and 3 instead of both system and cancer site as in Analysis 1.

No sample size was calculated before the analysis as we have used all the available data in all analyses. Details of the numbers of inclusion and exclusion criteria in each analysis are available in Fig. 1. No randomisation or blinding was used. All analyses were performed using SAS version 9.4 and R version 4.5.0.

### Reporting summary

Further information on research design is available in the Nature Portfolio Reporting Summary linked to this article.

## Data availability

The data used in this study can be made available upon request to the Research Data Office at Karolinska Institutet via rdo@ki.se after ensuring compliance with relevant legislation and General Data Protection Regulation. Source data for Figs. 2–5, Supplementary Figs. 1-2 and disaggregated data of sex were provided with this paper. Source data are provided with this paper.

## Code availability

The codes used for analyses in this study are publicly available at https://doi.org/10.6084/m9.figshare.32054538.

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

## Acknowledgements

The study was supported by ìShizu Matsumuraîs Donation (grant no. FS-2023:0003 to DW) and the Swedish Cancer Society (grant no. 23 2741Pj to FF). D.W. was supported by the International Postdoc Grant from the Swedish Research Council (grant no. 2023-00206). K.H. was supported by the Starting Grant for High-Level Talents in West China Hospital Sichuan University to (grant no. 137250062). The funders did not play any role in the study design, data collection and analysis, decision to publish, or preparation of the manuscript.

## Author contributions

D.W. and K.H conceived and designed the study. H.P., Q.W., and Q.L. contributed to the study design. H.P. and D.W. performed the literature search and performed data analyses. J.W., K.Y., X.Z., S.H.A, W.D., N.H., M.F., and F.F. contributed to data interpretation. H.P., K.H. and D.W. wrote the initial draft. All authors contributed to manuscript review and revision.

## Funding

## Competing interests

The authors declare no competing interests.
