## [Transparent Peer Review file · Nature Communications]

Metabolic and inflammatory biomarker trajectories after a cancer diagnosis and the risk of cardiovascular diseases

Corresponding Author: Dr Kejia Hu

Version 0:

Reviewer comments:

Reviewer #1

(Remarks to the Author)

ENCOMMS-25-49785

Title: Metabolic and inflammatory biomarker trajectories after a cancer diagnosis and the risk of cardiovascular diseases

General assessment:

This manuscript presents a large-scale, registry-based cohort study evaluating longitudinal trajectories of metabolic and inflammatory biomarkers in cancer patients and their association with subsequent cardiovascular disease (CVD) risk. Using the Swedish AMORIS cohort (n = 757,040; ~90,000 with cancer), the authors employed Poisson regression, linear mixed-effect models, and latent class growth modeling (LCGM) to demonstrate that cancer patients exhibit distinct biomarker trajectories—especially in glucose, albumin, and uric acid—that are significantly associated with increased CVD risk. These findings suggest the potential utility of routine blood biomarkers for long-term cardiovascular risk stratification in cancer survivors.

However, I have a few concerns that should be addressed prior to publication.

MAJOR COMMENTS:

1. Interpretation of biomarker trajectories and causal inference

The manuscript presents strong associations between certain biomarker trajectories (e.g., glucose, albumin, uric acid) and CVD risk. However, given the observational nature of the study and potential for reverse causality or residual confounding (e.g., cancer treatments, pre-existing cardiometabolic conditions), more emphasis should be placed on cautious interpretation. While the authors briefly mention these limitations, a more nuanced discussion of how cancer therapies (e.g., glucocorticoids, anthracyclines) or unmeasured variables (e.g., obesity, dyslipidemia treatment) may impact biomarker levels and CVD risk would strengthen the manuscript.

2. Clarification of trajectory classification and clinical relevance

The methodology for defining latent classes is rigorous, yet the clinical interpretability of some trajectories (e.g., “low-stable” vs. “moderate-stable” albumin) remains limited. The authors should elaborate on how such trajectories could be applied in clinical settings. Are cutoffs or change rates defined? Could thresholds be proposed to aid in implementation?

3. Inclusion of HbA1c and cancer treatment variables

To fully understand the rationale behind the biomarker selection and their interpretation, I found it necessary as a reviewer to revisit several prior studies conducted within the AMORIS cohort. Given the extensive use of AMORIS for metabolic and cardiovascular research, the authors are encouraged to more explicitly contextualize their approach within this existing body of work.

The manuscript focuses on glucose and fructosamine but does not explore HbA1c, a widely accepted long-term glycemic marker. While this may be due to data limitations, a brief justification is warranted. HbA1c remains clinically useful even in cancer patients, as shown in the GlicoOnco study (Toyoshima et al., Clinics, 2023), which demonstrated that both

fructosamine and HbA1c remained valid glycemic markers despite common oncologic confounders such as anemia, hypoproteinemia, and chemotherapy exposure.

The use of fructosamine as a short-term glycemic marker is well supported, especially in the AMORIS cohort. Malmström et al. (PLOS ONE, 2014) validated fructosamine's strong correlation with glucose and HbA1c, its diagnostic accuracy for diabetes, and its stability across fasting states. In the same cohort, Wulaningsih et al. (PLOS ONE, 2013) observed an inverse association between fructosamine and cancer risk, raising the hypothesis that fructosamine may reflect broader metabolic or nutritional states beyond glycemia. More recently, Connor et al. (NPJ Breast Cancer, 2019) found fructosamine to be a strong predictor of both breast cancer-specific and all-cause mortality, mediating nearly half of the diabetes-mortality relationship.

The current manuscript builds on this AMORIS-based literature by focusing on post-cancer trajectories and CVD outcomes. Given this background, the authors may wish to explicitly position their findings in relation to previous AMORIS studies, particularly where interpretations of fructosamine diverge. A brief discussion of these contrasting associations would enrich the physiological plausibility of their results.

Additionally, cancer treatment modalities—particularly chemotherapy, radiotherapy, corticosteroids, and hormone therapy—are known to influence both CVD risk and metabolic biomarkers, including fructosamine. Several studies (Psarakis, 2006; Hershey, 2017) have shown that oncologic treatments frequently induce hyperglycemia or catabolic states that can distort glycemic markers. Even if treatment data are not available in the current study, the authors should explicitly acknowledge this limitation and caution against overinterpreting the prognostic meaning of glycemic trajectories without accounting for treatment effects.

Finally, the authors may wish to clarify whether the goal of identifying these trajectories is prognostic (risk stratification) or mechanistic (suggesting causal pathways), and whether such latent classes can be applied clinically—e.g., through threshold-based classification or external validation.

Suggested References for Authors:

1. Toyoshima MTK et al. Fructosamine and HbA1c in hospitalized cancer patients: validity and clinical implications. *Clinics (Sao Paulo)*. 2023;78:100285.
2. Wulaningsih W et al. Low serum fructosamine and cancer risk: a population-based cohort study in AMORIS. *PLoS One*. 2013;8(9):e75211.
3. Malmström H et al. Fructosamine is a useful screening test for diabetes and glucose abnormalities: a study from the AMORIS cohort. *PLoS One*. 2014;9(10):e111463.
4. Connor AE et al. Fructosamine and cancer mortality in breast cancer survivors. *NPJ Breast Cancer*. 2019;5:1.
5. Psarakis HM. Clinical challenges in caring for patients with diabetes and cancer. *Diabetes Spectrum*. 2006;19(3):157–162.
6. Hershey DS. Importance of glycemic control in cancer patients with diabetes. *Asia Pac J Oncol Nurs*. 2017;4(4):313–317.

4. Generalizability and external validation

The study is based in Sweden and uses data from individuals undergoing health check-ups or outpatient laboratory monitoring. The generalizability to broader or more diverse populations (e.g., non-European ancestry, active treatment settings) or to lower-resource contexts may be limited. This limitation deserves clearer discussion.

5. Lack of stratified analysis or detailed description by cancer type

Although Supplementary Figure 1 provides IRRs for CVD by cancer site, the main text lacks sufficient detail on the distribution of cancer types among included patients, particularly in the trajectory and risk analyses. Given that cancer-related CVD risk and biomarker alterations are known to vary by tumor site and treatment (e.g., hematologic malignancies vs. breast or lung cancer), the authors should:

- Provide a table summarizing the most common cancer types among participants included in trajectory analyses;
- Discuss whether the identified biomarker trajectories and their associations with CVD risk varied by cancer type;
- Acknowledge the limitation of aggregating all cancers as a single exposure variable when presenting trajectory-based risk estimates.

6. Limited stratification by cancer type, despite partial acknowledgment

While the authors briefly acknowledge in the Discussion that they were unable to fully explore differences by cancer type, this limitation is only superficially addressed. The main analyses—including trajectory modeling and associations with cardiovascular outcomes—treat cancer as a single, aggregated exposure. Given the substantial heterogeneity in cardiovascular risk, treatment modalities, and metabolic response across different cancer types (e.g., hematological vs. breast vs. lung cancer), a more detailed description and analysis are warranted. At minimum, the manuscript should:

- Present a table describing the distribution of cancer types among included participants;
- Clarify whether trajectory patterns or risk associations differed meaningfully by cancer site;
- Explicitly discuss this limitation and its clinical implications in greater depth than currently provided.

7. Lack of treatment modality and intent data — insufficiently addressed

The authors briefly acknowledge the lack of treatment data in the Discussion, but this important limitation is not sufficiently explored. Cancer therapies—including chemotherapy, radiotherapy, and hormone therapy—can independently influence both biomarker levels (e.g., glucose, CRP, albumin) and cardiovascular risk. Additionally, the intent of treatment (curative vs. adjuvant vs. palliative) profoundly affects patients' metabolic and inflammatory profiles, life expectancy, and clinical

decision-making. The authors should:

- Clearly discuss how the absence of treatment information may confound or mediate observed associations;
- Emphasize that certain biomarker trajectories may reflect treatment effects rather than cancer biology alone;
- Suggest this as a critical direction for future research, particularly to tailor cardiovascular prevention strategies in oncology care.

The observed transient increase in leukocyte counts and LDL/HDL ratio shortly after cancer diagnosis, followed by normalization over time, strongly suggests a treatment-related effect. Cytotoxic chemotherapy, corticosteroids, and supportive agents (e.g., growth factors) are known to impact both inflammatory and lipid biomarkers. The lack of detailed treatment data limits the ability to distinguish whether these patterns reflect tumor biology, treatment effects, or recovery. A more explicit discussion of this temporal association would enhance the interpretability of these biomarker trajectories and support the call for stratification by treatment type and intent.

The interpretation of uric acid trajectories may also be confounded by cancer therapy. Cytotoxic treatments can increase purine turnover through tumor cell lysis, even in the absence of overt tumor lysis syndrome. In addition, renal toxicity, nutritional changes, and systemic inflammation may alter uric acid metabolism or excretion. Given that uric acid was one of the biomarkers showing modest but sustained differences in cancer patients, a discussion of these potential treatment-related effects would be appropriate.

8. Role of age as an effect modifier

While age was appropriately included as an adjustment variable and time scale in the models, the potential role of age as an effect modifier was not explored. Given the markedly higher relative risk of CVD observed in younger cancer patients (e.g., IRR 2.62 for those <18), and their longer life expectancy, it would be clinically meaningful to assess whether biomarker trajectories have differential implications by age group. At minimum, the authors should discuss this possibility in the manuscript and consider it as a direction for future stratified analyses.

9. Ambiguity in the definition of HDL and LDL

Throughout the manuscript, the terms “HDL” and “LDL” are used without specifying whether they refer to the cholesterol content (HDL-C, LDL-C) or to the number/concentration of lipoprotein particles. This distinction has important clinical and physiological implications, especially in cancer and inflammation contexts where particle composition and cholesterol content may diverge. The authors should clearly define which variable was measured and clarify the laboratory methods used to obtain these values.

10. Clinical applicability of latent trajectories and causality considerations

While the identification of biomarker trajectories using LCGM is methodologically robust, the manuscript would benefit from a clearer discussion of its potential application in clinical settings. Can these latent classes be used for patient stratification or decision-making? Are the trajectories reproducible across cohorts? Additionally, the distinction between causal inference and prognostic modeling should be made explicit—particularly whether these patterns could be used to guide preventive cardiovascular care in cancer survivors or are simply statistical markers of risk.

11. Lack of adjustment for statin use and lipid-lowering therapies

The manuscript analyzes trajectories of lipid-related biomarkers such as LDL-cholesterol, HDL-cholesterol, total cholesterol, and apolipoproteins, which are directly influenced by the use of statins and other lipid-lowering agents. However, the models do not appear to include adjustment for medication use—particularly statins—nor is this limitation discussed explicitly. This omission raises concerns regarding potential confounding, as statin therapy not only lowers LDL and ApoB levels, but also has pleiotropic effects that may impact cardiovascular outcomes independently of lipid levels. Given that cancer survivors may have differential access to preventive cardiovascular care, including statin use, the absence of this adjustment could bias the associations observed between lipid trajectories and cardiovascular disease. The authors should either clarify the availability of prescription data or acknowledge the lack of statin adjustment as a relevant limitation, especially in the context of interpreting lipid biomarker trajectories.

Minor Comments:

1. Abstract and Key Points: The abstract is clear but could specify whether the biomarkers were measured before or after cancer diagnosis in the LCGM analyses.
2. Terminology: Throughout the manuscript, clarify whether “albumin low-stable” refers to absolute hypoalbuminemia or relative to the population distribution.
3. Figure 3: Consider adding confidence intervals or number-at-risk per trajectory curve to improve interpretability.
4. Statistical methods: Clarify if the time-varying covariates (e.g., diabetes, psychiatric disorders) were updated at regular intervals or continuously tracked.
5. Supplementary Tables: Supplementary Table 6 is critical to the main findings and could be moved into the main text or better summarized in the Results.

(Remarks on code availability)

Reviewer #2

(Remarks to the Author)

Summary

This manuscript investigates the impact of a cancer diagnosis on metabolic and inflammatory biomarkers, and their trajectories over time. It also examines the association between these trajectories and the risk of subsequent cardiovascular events (CVD). Using data from the Amoris cohort and linked Swedish national health registers, the authors present a well-conducted and clearly written study on an important and timely topic. The analyses are appropriate, the presentation is clear, and the findings may have clinical and public health relevance. I have some minor comments that may assist in strengthening the manuscript.

Recommendation

Minor revisions (if data for sensitivity analyses are not available).

Comments

Definition of the Primary Outcome

o The manuscript refers to “cardiovascular disease” as the primary outcome. Based on the data sources used (e.g., inpatient register, cause of death registry), the outcome is likely restricted to major cardiovascular events requiring hospitalisation. If this is the case, please consider clarifying and/or modifying the terminology throughout the manuscript to avoid readers misinterpreting this as encompassing all CVD, including conditions that may be managed in primary care.

Potential for Missed Outcomes

o Would the addition of medication prescription data results have improved case ascertainment? Is this possible for these analyses?

Cancer Treatment-Specific Effects

o Different cancer treatments (e.g., cardiotoxic chemotherapies, radiotherapy that has damaged heart muscle) can influence CVD risk. If available, a sensitivity analysis stratified by treatment type would strengthen the study. If not feasible, please acknowledge this limitation explicitly.

Healthy Volunteer Bias

o As this is a secondary analysis of data from the Amoris cohort, it would be valuable to include discussion of the potential for healthy volunteer bias and/or other selection bias.

Residual Confounding

o Given the observational design and constraints of the available data, residual confounding is likely. The discussion could more clearly convey that this is “more likely than not” rather than simply possible.

Conclusion

Given the limitations of the study, the conclusion is somewhat strongly worded. Monitoring of blood biomarkers may be useful but further research is most certainly required before confirming this.

Minor Comments

Introduction (First Sentence)

o The statement on advances in early detection and treatment increasing survival could be rephrased for accuracy. For example: “Advances in early detection and cancer treatment have significantly increased survival rates for many cancer patients.”

Other

o Ensure consistent terminology for the primary outcome and biomarker names across the abstract, methods, results, and discussion.

(Remarks on code availability)

Version 1:

Reviewer comments:

Reviewer #1

(Remarks to the Author)

The authors have adequately addressed the major concerns raised in the previous round of peer review. Specifically:

- They now explicitly acknowledge the lack of data on cancer treatment modalities and recognize this as a major limitation that could influence biomarker trajectories and cardiovascular risk.
- They have clarified the short-term validity and use of fructosamine as a glycemic biomarker, and added discussion regarding its clinical relevance in oncology.
- The transient increase in leukocytes and LDL/HDL ratio following cancer diagnosis is now correctly attributed to possible treatment-related effects.
- The absence of statin use and other cardiometabolic medications has been added as a limitation.
- The authors also clarified that their findings are primarily prognostic in nature, not necessarily causal.

While the manuscript could further benefit from an explicit comparison to prior AMORIS-based studies (e.g., Wulaningsih et al., 2013; Malmström et al., 2014) and from a more detailed discussion on the clinical applicability of latent trajectory modeling, these are secondary issues that do not detract significantly from the overall strength of the study.

The manuscript is methodologically sound, clinically relevant, and provides valuable insights into longitudinal biomarker patterns and cardiovascular risk in cancer survivors.

I believe the manuscript is now suitable for acceptance in its current form or with minor editorial adjustments.

(Remarks on code availability)

Reviewer #3

(Remarks to the Author)

Thank you for the opportunity to review this manuscript. The authors have done a very good job of responding to the thoughtful and extensive comments raised by the reviewers so far. I had a few suggestions to improve the clarity of the work.

The results show that the risk increase of CVD was highest if the cancer was diagnosed before age 18. It would be useful to explicitly state that this was based on 60 cancers in that age group.

The part in the discussion about age could be written more clearly. Their finding does seem to fit with previous work (childhood cancer patients diagnosed before 18 years have been shown to experience approximately two- to three-fold greater CVD risk) but this is not clearly compared. The statement "It would be clinically meaningful to examine whether post-cancer biomarker trajectories differ by age at diagnosis and whether their implications vary across age groups. However, we were unable to do so due to lack of data on childhood cancer patients since the included participants were diagnosed with cancer at a mean age of 60.7" is not very clear. I think what the authors mean is that the numbers were too small in some age groups to support detailed analysis by age at diagnosis.

I find the inclusion of both organ system and site of cancer a little confusing, as the results switch from one to the other. It would be good to explicitly state that Analysis 1 is broken down by site, but due to numbers Analysis 2 and 3 could only be broken down by system (if that was the case). Or perhaps just stick with organ system throughout?

More of an editorial comment, but there are some very long paragraphs in the discussion and I would recommend breaking some of these up to aid readability.

(Remarks on code availability)

REVIEWER COMMENTS

Reviewer #1 (Remarks to the Author):

ENCOMMS-25-49785

Title: Metabolic and inflammatory biomarker trajectories after a cancer diagnosis and the risk of cardiovascular diseases

General assessment:

This manuscript presents a large-scale, registry-based cohort study evaluating longitudinal trajectories of metabolic and inflammatory biomarkers in cancer patients and their association with subsequent cardiovascular disease (CVD) risk. Using the Swedish AMORIS cohort (n = 757,040; ~90,000 with cancer), the authors employed Poisson regression, linear mixed-effect models, and latent class growth modeling (LCGM) to demonstrate that cancer patients exhibit distinct biomarker trajectories—especially in glucose, albumin, and uric acid—that are significantly associated with increased CVD risk. These findings suggest the potential utility of routine blood biomarkers for long-term cardiovascular risk stratification in cancer survivors.

However, I have a few concerns that should be addressed prior to publication.

Response: Thank you for the positive evaluation of our manuscript. We have addressed your specific concerns point-by-point and revised the manuscript accordingly. The page and line numbers of the modified text refer to the clean version of the manuscript.

MAJOR COMMENTS:

1. Interpretation of biomarker trajectories and causal inference

The manuscript presents strong associations between certain biomarker trajectories (e.g., glucose, albumin, uric acid) and CVD risk. However, given the observational nature of the study and potential for reverse causality or residual confounding (e.g., cancer treatments, pre-existing cardiometabolic conditions), more emphasis should be placed on cautious interpretation. While the authors briefly mention these limitations, a more nuanced discussion of how cancer therapies (e.g., glucocorticoids, anthracyclines) or unmeasured variables (e.g., obesity, dyslipidemia treatment) may impact biomarker levels and CVD risk would strengthen the manuscript.

Response: Thank you for the suggestion. We have now extended the discussion to include the potential roles of cancer therapies and unmeasured variables.

Modified text:

“Cancer therapies are known to influence both CVD risk and metabolic biomarkers. For example, cardiotoxicity is a concern for many chemotherapies, especially anthracyclines and certain targeted agents (e.g., Trastuzumab and Tyrosine Kinase Inhibitors) ²⁵. Chemotherapy, radiotherapy and hormone deprivation therapies have also been suggested to be associated

with insulin resistance, dyslipidemia and hyperglycemia²⁶⁻²⁹. Cytotoxic treatments can increase purine turnover through tumor cell lysis, even in the absence of overt tumor lysis syndrome³⁰. In addition, renal toxicity, nutritional changes, and systemic inflammation may alter metabolism or excretion of metabolic biomarkers^{31,32}. All of these might have contributed to the altered biomarker levels observed in cancer patients compared to non-cancer individuals in the present study, although the biomarker levels generally remained within the normal ranges. Nonetheless, as our trajectory analyses of biomarkers focused on long-term changes spanning across 12 years after a cancer diagnosis (Supplementary Figure 2), the potential influence of cancer therapies is limited.” (Discussion section, pages 7-8, lines 199-211)

“Altered biomarker levels may represent underlying health conditions and are manageable through corresponding treatments. We adjusted for psychiatric disorders and diabetes in the analyses as they were closely related to both the studied biomarkers and risk of CVD. Obesity is also a potential confounder; however, we had limited data on body mass index in the AMORIS cohort. Nevertheless, we speculate that some of these health conditions might be secondary to cancer diagnosis and could thus serve as mediators, rather than confounders, in the association between post-cancer biomarker trajectories and CVD risk, i.e., should not be adjusted for. Treatments for dyslipidemia, dysglycemia and hyperuricemia might, on the other hand, have led to an underestimation of the true association. As the treatments are aiming to modulate the altered biomarker levels to normal, i.e., some cancer patients with abnormal biomarker levels could appear as with “normal” biomarker levels thanks to the treatments. This might have contributed to the null association observed in Analysis 3 for lipids and lipid proteins. Nevertheless, we still found a higher risk of CVD among individuals with persistently high glucose levels. A critical direction for future research is clinical implementation of these biomarker trajectories, taken into consideration cancer therapies, comorbidities and treatment, to tailor cardiovascular prevention strategies in oncology care.” (Discussion section, page 8, lines 217-232)

2. Clarification of trajectory classification and clinical relevance

The methodology for defining latent classes is rigorous, yet the clinical interpretability of some trajectories (e.g., “low-stable” vs. “moderate-stable” albumin) remains limited. The authors should elaborate on how such trajectories could be applied in clinical settings. Are cutoffs or change rates defined? Could thresholds be proposed to aid in implementation?

Response: Thank you for the good comment. In the analysis of biomarker trajectories using latent class growth modeling, classes were derived through maximum likelihood estimation in the best-fit model for each biomarker. Model selection was guided by the lowest BIC, statistical significance of all polynomial terms ($P < 0.05$), and a group membership probability threshold of at least 5%. No clinical biomarker cutoffs were used to define class membership; instead, classification was based on the patterns of biomarker evolution over time.

Therefore, instead of relying on specific cutoffs, changes over time with a specific range may help guide clinical surveillance. For example, cancer patients with persistently rising glucose levels within the range of 8-15 mmol/L may face an increased risk of cardiovascular diseases. We have added more discussion about clinical implementation.

Modified text:

“The purpose of this study is not to develop clinical guidelines but rather explore if biomarker trajectories can be a promising area for further development of such guidelines. Further investigation, including mechanistic studies and validation in external clinical cohorts are needed before guiding clinical utility. In practice, fine mapping the temporal patterns of these biomarkers may assist in stratifying patients for intensified cardiovascular surveillance and, ultimately, prevention of CVD in the oncology setting.” (Discussion section, page 7, lines 185-191)

“We found that the group of cancer patients with constantly low albumin levels had a 56% higher risk of CVD than those with constantly moderate albumin levels. This finding is in line with existing literature showing that low albumin levels are associated with an increased CVD risk^{43,44}, suggesting that we could identify cancer patients at high risk of CVD by different levels of albumin even within the normal range of the general population (35-55 g/L)²⁰.” (Discussion section, pages 9-10, lines 266-270)

3. Inclusion of HbA1c and cancer treatment variables

To fully understand the rationale behind the biomarker selection and their interpretation, I found it necessary as a reviewer to revisit several prior studies conducted within the AMORIS cohort. Given the extensive use of AMORIS for metabolic and cardiovascular research, the authors are encouraged to more explicitly contextualize their approach within this existing body of work.

The manuscript focuses on glucose and fructosamine but does not explore HbA1c, a widely accepted long-term glycemic marker. While this may be due to data limitations, a brief justification is warranted. HbA1c remains clinically useful even in cancer patients, as shown in the GlicoOnco study (Toyoshima et al., Clinics, 2023), which demonstrated that both fructosamine and HbA1c remained valid glycemic markers despite common oncologic confounders such as anemia, hypoproteinemia, and chemotherapy exposure.

Response: The reviewer is correct that we did not include HbA1c in the analysis because of data limitation, i.e., HbA1c was only available for 2.3% of the cohort participants. We have added this as a limitation in the Discussion section.

Modified text:

“Despite being a widely recognized marker of long-term glycemic control, data on HbA1c was available for only 2.3% of the AMORIS participants who have prevalent or suspected diabetes³⁴ and was thus not included in our analysis. Future studies should incorporate HbA1c, given its strong correlation with fructosamine and glucose in cancer patients despite common oncologic confounders³⁵.”. (Discussion section, page 9, lines 236-240)

The use of fructosamine as a short-term glycemic marker is well supported, especially in the AMORIS cohort. Malmström et al. (PLOS ONE, 2014) validated fructosamine’s strong correlation with glucose and HbA1c, its diagnostic accuracy for diabetes, and its stability across fasting states. In the same cohort, Wulaningsih et al. (PLOS ONE, 2013) observed an inverse association between fructosamine and cancer risk, raising the hypothesis that fructosamine may reflect broader metabolic or nutritional states beyond glycemia. More

recently, Connor et al. (NPJ Breast Cancer, 2019) found fructosamine to be a strong predictor of both breast cancer–specific and all-cause mortality, mediating nearly half of the diabetes–mortality relationship.

The current manuscript builds on this AMORIS-based literature by focusing on post-cancer trajectories and CVD outcomes. Given this background, the authors may wish to explicitly position their findings in relation to previous AMORIS studies, particularly where interpretations of fructosamine diverge. A brief discussion of these contrasting associations would enrich the physiological plausibility of their results.

Response: Thank for the good suggestion. We have added discussion about the interpretation and comparison of existing studies on fructosamine.

Modified text:

“Evidence has accumulated to support fructosamine as a glycemic marker in the AMORIS cohort ³⁶. Mean of 4 repeated measurements of fructosamine after enrollment to AMORIS, before or after cancer diagnosis, was associated with a reduced cancer risk ³⁷, while a single measurement of fructosamine around 5 months post-diagnosis was linked to an increased risk of death among breast cancer patients ³⁸. These conflicting findings may suggest that elevated fructosamine levels reflect different underlying metabolic or nutritional states, although both studies have relied on single value of fructosamine. In contrast, we investigated longitudinal patterns of this biomarker following cancer diagnosis and found that, compared to cancer-free individuals, cancer patients showed consistently high levels during the 12 years after cancer diagnosis. However, we did not identify multiple latent classes of fructosamine trajectories among cancer patients, indicating that most patients followed a similar pattern. To the best of our knowledge, AMORIS is one of the largest cohorts worldwide regarding measurements of fructosamine. The absence of distinct trajectories may be due to our aggregated analysis across all cancer types, which could mask cancer-specific physiological variations..” (Discussion section, page 9, lines 241-254)

Additionally, cancer treatment modalities—particularly chemotherapy, radiotherapy, corticosteroids, and hormone therapy—are known to influence both CVD risk and metabolic biomarkers, including fructosamine. Several studies (Psarakis, 2006; Hershey, 2017) have shown that oncologic treatments frequently induce hyperglycemia or catabolic states that can distort glycemic markers. Even if treatment data are not available in the current study, the authors should explicitly acknowledge this limitation and caution against overinterpreting the prognostic meaning of glycemic trajectories without accounting for treatment effects.

Response: Thank you for the good comment. We have added discussion about cancer treatment.

Modified text:

“Cancer therapies are known to influence both CVD risk and metabolic biomarkers. For example, cardiotoxicity is a concern for many chemotherapies, especially anthracyclines and certain targeted agents (e.g., Trastuzumab and Tyrosine Kinase Inhibitors) ²⁵. Chemotherapy, radiotherapy and hormone deprivation therapies have also been suggested to be associated with insulin resistance, dyslipidemia and hyperglycemia ^{26–29}. Cytotoxic treatments can increase purine turnover through tumor cell lysis, even in the absence of overt tumor lysis

syndrome³⁰. In addition, renal toxicity, nutritional changes, and systemic inflammation may alter metabolism or excretion of metabolic biomarkers^{31,32}. All of these might have contributed to the altered biomarker levels observed in cancer patients compared to non-cancer individuals in the present study, although the biomarker levels generally remained within the normal ranges. Nonetheless, as our trajectory analyses of biomarkers focused on long-term changes spanning across 12 years after a cancer diagnosis (Supplementary Figure 2), the potential influence of cancer therapies is limited.” (Discussion section, pages 7-8, lines 199-211)

“Third, although multiple covariates were adjusted for in the analyses, there may still be residual confounding from unknown or unmeasured factors, e.g., cancer therapies and sub-clinical cardiometabolic conditions not attended by specialist care, like atherosclerosis, dyslipidemia, dysglycemia, hyperuricemia, and their treatments. Therefore, caution is warranted when interpreting our findings and considering the clinical utility.” (Discussion section, pages 10, lines 290-295)

Finally, the authors may wish to clarify whether the goal of identifying these trajectories is prognostic (risk stratification) or mechanistic (suggesting causal pathways), and whether such latent classes can be applied clinically—e.g., through threshold-based classification or external validation.

Response: Thanks for the opportunity to clarify. Our aim was to take advantage of the AMORIS biomarker data to identify who is at higher or lower risk of CVD among cancer patients. Therefore, we think prognostic (risk stratification) is more relevant in our study. Although we have tried our best to adjust for potential confounders, residual confounding can't be completely ruled out due to the lack of information, such as genetic and lifestyle factors. Therefore, we could not infer causal pathways. Regardless, our findings call for further mechanistic demonstration and independent validation for these biomarker trajectories, and may ultimately contribute to the development of clinical prevention guidelines.

We are of course willing to reconsider our position should the reviewer think differently.

Modified text:

“Third, although multiple covariates were adjusted for in the analyses, there may still be residual confounding from unknown or unmeasured factors, e.g., cancer therapies and sub-clinical cardiometabolic conditions not attended by specialist care, like atherosclerosis, dyslipidemia, dysglycemia, hyperuricemia, and their treatments. Therefore, caution is warranted when interpreting our findings and considering the clinical utility.” (Discussion section, pages 10, lines 290-295)

“The purpose of this study is not to develop clinical guidelines but rather explore if biomarker trajectories can be a promising area for further development of such guidelines. Further investigation, including mechanistic studies and validation in external clinical cohorts are needed before guiding clinical utility. In practice, fine mapping the temporal patterns of these biomarkers may assist in stratifying patients for intensified cardiovascular surveillance and, ultimately, prevention of CVD in the oncology setting.” (Discussion section, page 7, lines

185-191)

Suggested References for Authors:

1. Toyoshima MTK et al. Fructosamine and HbA1c in hospitalized cancer patients: validity and clinical implications. *Clinics (Sao Paulo)*. 2023;78:100285.
2. Wulaningsih W et al. Low serum fructosamine and cancer risk: a population-based cohort study in AMORIS. *PLoS One*. 2013;8(9):e75211.
3. Malmström H et al. Fructosamine is a useful screening test for diabetes and glucose abnormalities: a study from the AMORIS cohort. *PLoS One*. 2014;9(10):e111463.
4. Connor AE et al. Fructosamine and cancer mortality in breast cancer survivors. *NPJ Breast Cancer*. 2019;5:1.
5. Psarakis HM. Clinical challenges in caring for patients with diabetes and cancer. *Diabetes Spectrum*. 2006;19(3):157–162.
6. Hershey DS. Importance of glycemic control in cancer patients with diabetes. *Asia Pac J Oncol Nurs*. 2017;4(4):313–317.

Response: Thank you so much for the references. We have cited these papers in the updated discussion.

4. Generalizability and external validation

The study is based in Sweden and uses data from individuals undergoing health check-ups or outpatient laboratory monitoring. The generalizability to broader or more diverse populations (e.g., non-European ancestry, active treatment settings) or to lower-resource contexts may be limited. This limitation deserves clearer discussion.

Response: We agree with the reviewer about the generalizability and have expanded this in the discussion section.

Modified text:

“Second, this study was based on individuals who underwent occupational health check-ups or outpatient laboratory tests in Stockholm, Sweden. Although this does not affect the internal validity of studies derived from the AMORIS cohort, the generalizability of our findings to broader or more diverse populations (e.g., those of non-European ancestry or individuals in lower-resource contexts) may be limited. Further research in other populations and settings is warranted to validate our findings.” (Discussion section, pages 10, lines 284-290)

5. Lack of stratified analysis or detailed description by cancer type

Although Supplementary Figure 1 provides IRRs for CVD by cancer site, the main text lacks sufficient detail on the distribution of cancer types among included patients, particularly in the trajectory and risk analyses. Given that cancer-related CVD risk and biomarker alterations are known to vary by tumor site and treatment (e.g., hematologic malignancies vs. breast or lung cancer), the authors should:

- Provide a table summarizing the most common cancer types among participants included in trajectory analyses;
- Discuss whether the identified biomarker trajectories and their associations with CVD risk varied by cancer type;

- Acknowledge the limitation of aggregating all cancers as a single exposure variable when presenting trajectory-based risk estimates.

Response: As suggested, we have:

1) Provided tables summarizing the most common cancer types among participants included in Analyses 1-3 (new Supplementary Tables 3, 4, and 6). The common cancer types among individuals included in the three analyses were largely consistent, with cancers of the breast and reproductive system being the most frequent. We have reported the distribution of cancer types among cancer patients in the Results section and Supplementary materials accordingly.

Modified text:

“In the AMORIS cohort, the three most common cancer types by organ system were those affecting the breast and reproductive system, the digestive system, and the hematopoietic system, while the three most common site-specific cancers were breast cancer, colorectal cancer, and melanoma (Supplementary Table 3).” (Results section, Analysis 1, page 4, lines 100-103)

“In this analysis, the top three cancer types by organ system were cancers in the breast and reproductive system, the digestive system, and the urinary system (Supplementary Table 4).” (Results section, Analysis 2, page 5, lines 111-112)

“A total of 2206 individuals were included in Analysis 3, among whom the three most frequent cancer types by organ system were the same as those in Analysis 2 (Supplementary Table 6).” (Results section, Analysis 3, page 5, lines 120-121)

2) Discussed potential variations in biomarker trajectories after cancer diagnosis and their associations with CVD risk according to cancer types.

Modified text:

“Nonetheless, as our trajectory analyses of biomarkers focused on long-term changes spanning across 12 years after a cancer diagnosis (Supplementary Figure 2), the potential influence of cancer therapies is limited. Moreover, the observed associations between latent classes of longitudinal trajectories biomarker after diagnosis and CVD risk appear to be largely consistent across the three most frequent cancer types by organ systems for some biomarkers (e.g., glucose), though not for others (e.g., albumin) (Supplementary Table 9). Further large-scale studies are warranted to confirm the robustness of these associations and to clarify potential sources of heterogeneity.” (Discussion section, page 8, lines 209-216)

3) Acknowledged the limitation of aggregating all cancers as a single exposure variable when presenting trajectory-based risk estimates and its clinical implications.

Modified text:

“Fourth, due to the limit of sample size, we studied cancer as one single category in Analyses 2 and 3, but it is important to keep in mind that cancer is largely heterogeneous in terms of etiology, pathology, treatment and metabolic response. Although we were able to grasp

some universal metabolic changes in all cancer types ⁴⁶, future study should extend the effort to specific cancer types and explore the underlying mechanism in the context of each cancer before clinical implementation.” (Discussion section, page 10, lines 295-300)

6. Limited stratification by cancer type, despite partial acknowledgment

While the authors briefly acknowledge in the Discussion that they were unable to fully explore differences by cancer type, this limitation is only superficially addressed. The main analyses—including trajectory modeling and associations with cardiovascular outcomes—treat cancer as a single, aggregated exposure. Given the substantial heterogeneity in cardiovascular risk, treatment modalities, and metabolic response across different cancer types (e.g., hematological vs. breast vs. lung cancer), a more detailed description and analysis are warranted. At minimum, the manuscript should:

- Present a table describing the distribution of cancer types among included participants;
- Clarify whether trajectory patterns or risk associations differed meaningfully by cancer site;
- Explicitly discuss this limitation and its clinical implications in greater depth than currently provided.

Response: Please see the response to comment 5.

7. Lack of treatment modality and intent data — insufficiently addressed

The authors briefly acknowledge the lack of treatment data in the Discussion, but this important limitation is not sufficiently explored. Cancer therapies—including chemotherapy, radiotherapy, and hormone therapy—can independently influence both biomarker levels (e.g., glucose, CRP, albumin) and cardiovascular risk. Additionally, the intent of treatment (curative vs. adjuvant vs. palliative) profoundly affects patients’ metabolic and inflammatory profiles, life expectancy, and clinical decision-making. The authors should:

- Clearly discuss how the absence of treatment information may confound or mediate observed associations;
- Emphasize that certain biomarker trajectories may reflect treatment effects rather than cancer biology alone;
- Suggest this as a critical direction for future research, particularly to tailor cardiovascular prevention strategies in oncology care.

The observed transient increase in leukocyte counts and LDL/HDL ratio shortly after cancer diagnosis, followed by normalization over time, strongly suggests a treatment-related effect. Cytotoxic chemotherapy, corticosteroids, and supportive agents (e.g., growth factors) are known to impact both inflammatory and lipid biomarkers. The lack of detailed treatment data limits the ability to distinguish whether these patterns reflect tumor biology, treatment effects, or recovery. A more explicit discussion of this temporal association would enhance the interpretability of these biomarker trajectories and support the call for stratification by treatment type and intent.

The interpretation of uric acid trajectories may also be confounded by cancer therapy. Cytotoxic treatments can increase purine turnover through tumor cell lysis, even in the absence of overt tumor lysis syndrome. In addition, renal toxicity, nutritional changes, and systemic inflammation may alter uric acid metabolism or excretion. Given that uric acid was one of the biomarkers showing modest but sustained differences in cancer patients, a discussion of these potential treatment-related effects would be appropriate.

Response:

We agree that cancer treatment can influence the regulation of inflammation and lipid metabolism in the short term after diagnosis. However, the elevations in leukocyte counts and LDL/HDL ratio did not peak until approximately five years after diagnosis. Previous evidence has shown that LDL levels increase significantly one month after chemotherapy but gradually return towards baseline within 12 months (Xu et al. 2020). Therefore, we believe that the long-term temporal patterns of biomarkers—spanning up to 12 years—between cancer patients and non-cancer individuals are largely driven by underlying biological differences rather than treatment effects.

Modified text:

“Cancer therapies are known to influence both CVD risk and metabolic biomarkers. For example, cardiotoxicity is a concern for many chemotherapies, especially anthracyclines and certain targeted agents (e.g., Trastuzumab and Tyrosine Kinase Inhibitors)²⁵. Chemotherapy, radiotherapy and hormone deprivation therapies have also been suggested to be associated with insulin resistance, dyslipidemia and hyperglycemia^{26–29}. Cytotoxic treatments can increase purine turnover through tumor cell lysis, even in the absence of overt tumor lysis syndrome³⁰. In addition, renal toxicity, nutritional changes, and systemic inflammation may alter metabolism or excretion of metabolic biomarkers^{31,32}. All of these might have contributed to the altered biomarker levels observed in cancer patients compared to non-cancer individuals in the present study, although the biomarker levels generally remained within the normal ranges. Nonetheless, as our trajectory analyses of biomarkers focused on long-term changes spanning across 12 years after a cancer diagnosis (Supplementary Figure 2), the potential influence of cancer therapies is limited. Moreover, the observed associations between latent classes of longitudinal trajectories biomarker after diagnosis and CVD risk appear to be largely consistent across the three most frequent cancer types by organ systems for some biomarkers (e.g., glucose), though not for others (e.g., albumin) (Supplementary Table 9). Further large-scale studies are warranted to confirm the robustness of these associations and to clarify potential sources of heterogeneity.

Altered biomarker levels may represent underlying health conditions and are manageable through corresponding treatments. We adjusted for psychiatric disorders and diabetes in the analyses as they were closely related to both the studied biomarkers and risk of CVD. Obesity is also a potential confounder; however, we had limited data on body mass index in the AMORIS cohort. Nevertheless, we speculate that some of these health conditions might be secondary to cancer diagnosis and could thus serve as mediators, rather than confounders, in the association between post-cancer biomarker trajectories and CVD risk, i.e., should not be adjusted for. Treatments for dyslipidemia, dysglycemia and hyperuricemia might, on the other hand, have led to an underestimation of the true association. As the treatments are aiming to modulate the altered biomarker levels to normal, i.e., some cancer patients with abnormal biomarker levels could appear as with “normal” biomarker levels thanks to the treatments. This might have contributed to the null association observed in Analysis 3 for lipids and lipid proteins. Nevertheless, we still found a higher risk of CVD among individuals with persistently high glucose levels. A critical direction for future research is clinical implementation of these biomarker trajectories, taken into consideration cancer therapies,

comorbidities and treatment, to tailor cardiovascular prevention strategies in oncology care.” (Discussion section, pages 7-8, lines 199-232)

8. Role of age as an effect modifier

While age was appropriately included as an adjustment variable and time scale in the models, the potential role of age as an effect modifier was not explored. Given the markedly higher relative risk of CVD observed in younger cancer patients (e.g., IRR 2.62 for those <18), and their longer life expectancy, it would be clinically meaningful to assess whether biomarker trajectories have differential implications by age group. At minimum, the authors should discuss this possibility in the manuscript and consider it as a direction for future stratified analyses.

Response: Since the mean age at cancer diagnosis was 60.7 among the cancer patients with ≥ 3 measurements of biomarkers in AMORIS cohort, we unfortunately lacked biomarker data after diagnosis for childhood cancer patients (age at diagnosis <18 years) and were therefore unable to examine whether biomarker trajectories differ by age at diagnosis. We agree that, given the high relative risk of CVD in young cancer patients observed in both the present study and previous studies (Gudmundsdottir et al. 2015; Winther et al. 2018) the predictive value of biomarker trajectories for CVD in this group may be considerable. Future studies are warranted to investigate these associations in younger populations. We have now added corresponding discussion in the manuscript.

Modified text:

“Given that childhood cancer patients (e.g., diagnosed before 18 years) have been shown to experience a substantially higher risk of CVD after cancer diagnosis—approximately two- to three-fold greater than that of cancer-free individuals^{23,24} - and have a longer life expectancy, it would be clinically meaningful to examine whether post-cancer biomarker trajectories differ by age at diagnosis and whether their implications vary across age groups. However, we were unable to do so due to lack of data on childhood cancer patients since the included participants were diagnosed with cancer at a mean age of 60.7.” (Discussion section, page 7, lines 191-198)

9. Ambiguity in the definition of HDL and LDL

Throughout the manuscript, the terms “HDL” and “LDL” are used without specifying whether they refer to the cholesterol content (HDL-C, LDL-C) or to the number/concentration of lipoprotein particles. This distinction has important clinical and physiological implications, especially in cancer and inflammation contexts where particle composition and cholesterol content may diverge. The authors should clearly define which variable was measured and clarify the laboratory methods used to obtain these values.

Response: Concentrations of HDL-cholesterol (HDL-C) and LDL-cholesterol (LDL-C) were measured in AMORIS cohort (Walldius et al. 2001). We have clarified the measurement of ApoB, ApoA-I, TC, TG, HDL-C, and LDL-C in the Methods section.

Modified text:

“AMORIS Participants had their first complete profile of ApoB, ApoA-I, TC and TG done simultaneously, and the ApoB/ApoA-I ratio, LDL-cholesterol, and HDL-cholesterol, were then

calculated accordingly ⁵¹. Apolipoproteins were measured with immunoturbidimetry. Concentrations of total cholesterol and triglycerides by enzymatic techniques. These methods were fully automated and were described previously ^{52,53}. Concentrations of LDL-cholesterol and HDL-cholesterol were calculated by the Jungner formula: $LDL=0.48+0.99TC-0.23TG-1.58ApoA1$; $HDL=TC-0.45TG-LDL$ for around 85% of the participants. The rest 15% had their LDL calculated by Friedewald formula ⁵⁴ and HDL measured directly from the blood by an automated precipitation method (Boehringer Mannheim GmbH, Mannheim, Germany) ⁵⁵." (Methods section, page 12, lines 352-361)

10. Clinical applicability of latent trajectories and causality considerations

While the identification of biomarker trajectories using LCGM is methodologically robust, the manuscript would benefit from a clearer discussion of its potential application in clinical settings. Can these latent classes be used for patient stratification or decision-making? Are the trajectories reproducible across cohorts? Additionally, the distinction between causal inference and prognostic modeling should be made explicit—particularly whether these patterns could be used to guide preventive cardiovascular care in cancer survivors or are simply statistical markers of risk.

Response: Please see the response to the last point of your comment 3, starting with "Finally".

11. Lack of adjustment for statin use and lipid-lowering therapies

The manuscript analyzes trajectories of lipid-related biomarkers such as LDL-cholesterol, HDL-cholesterol, total cholesterol, and apolipoproteins, which are directly influenced by the use of statins and other lipid-lowering agents. However, the models do not appear to include adjustment for medication use—particularly statins—nor is this limitation discussed explicitly. This omission raises concerns regarding potential confounding, as statin therapy not only lowers LDL and ApoB levels, but also has pleiotropic effects that may impact cardiovascular outcomes independently of lipid levels. Given that cancer survivors may have differential access to preventive cardiovascular care, including statin use, the absence of this adjustment could bias the associations observed between lipid trajectories and cardiovascular disease. The authors should either clarify the availability of prescription data or acknowledge the lack of statin adjustment as a relevant limitation, especially in the context of interpreting lipid biomarker trajectories.

Response: We agree with the reviewer and have acknowledged the limitation of lack of adjustment for preventive cardiovascular medication use. However, we did not adjust for statin use due to below reasons:

1. The Swedish Prescribed Drug Register was established since 2005, and we were unable to acquire data of statin use before that.
2. In Sweden, statins were first approved and marketed in 1988 (Hajar 2011), but the utilization remained low until 2000 (16.8 DDD per 1,000 inhabitants per day) (Treciokiene et al. 2024). Cancer diagnoses and biomarker measurements were made during 1985-1996, a period when statins were not widely used.

Modified text:

“Fifth, attendance to preventive cardiovascular care, including statin use, might be a potential confounder given that cancer survivors may have differential access to it. However, we were unable to find a suitable variable for adjustment. Data on prescribed statin use from the Swedish Prescribed Drug Register have been available only since 2005, which limits the ability to use medication prescription data to identify milder CVD as the outcome of interest. Moreover, statins were not commonly used during 1985-1996, the period in which cancer diagnoses and biomarker measurements were made in our study^{47,48}.” (Discussion section, pages 10-11, lines 300-307)

“Altered biomarker levels may represent underlying health conditions and are manageable through corresponding treatments. We adjusted for psychiatric disorders and diabetes in the analyses as they were closely related to both the studied biomarkers and risk of CVD. Obesity is also a potential confounder; however, we had limited data on body mass index in the AMORIS cohort. Nevertheless, we speculate that some of these health conditions might be secondary to cancer diagnosis and could thus serve as mediators, rather than confounders, in the association between post-cancer biomarker trajectories and CVD risk, i.e., should not be adjusted for. Treatments for dyslipidemia, dysglycemia and hyperuricemia might, on the other hand, have led to an underestimation of the true association. As the treatments are aiming to modulate the altered biomarker levels to normal, i.e., some cancer patients with abnormal biomarker levels could appear as with “normal” biomarker levels thanks to the treatments. This might have contributed to the null association observed in Analysis 3 for lipids and lipid proteins. Nevertheless, we still found a higher risk of CVD among individuals with persistently high glucose levels. A critical direction for future research is clinical implementation of these biomarker trajectories, taken into consideration cancer therapies, comorbidities and treatment, to tailor cardiovascular prevention strategies in oncology care.” (Discussion section, page 8, lines 217-232)

Minor Comments:

1. Abstract and Key Points: The abstract is clear but could specify whether the biomarkers were measured before or after cancer diagnosis in the LCGM analyses.

Response: We have clarified in the Abstract that LCGM was used to identify biomarker trajectories after cancer diagnosis.

Modified text:

“We identified biomarker trajectories after cancer diagnosis for 11 biomarkers using latent class growth modelling.” (Abstract section, page 2, lines 34-35)

2. Terminology: Throughout the manuscript, clarify whether "albumin low-stable" refers to absolute hypoalbuminemia or relative to the population distribution.

Response: The three trajectories of albumin, i.e., low-, moderate-, and high-stable levels, were also within the normal range (35-55 g/L). The trajectory of "albumin low-stable" refers to a relatively low and stable level within 35–39 g/L. We have clarified this throughout the manuscript.

3. Figure 3: Consider adding confidence intervals or number-at-risk per trajectory curve to improve interpretability.

Response: In both Figures 3 and 4, we have presented confidence intervals and reported the number of individuals included in the analysis of each biomarker. We have now clarified it in the legend of Figure 3 and 4.

Modified text:

“Shaded areas represent 95% confidence intervals.” (Page 23, lines 660, 678)

4. Statistical methods: Clarify if the time-varying covariates (e.g., diabetes, psychiatric disorders) were updated at regular intervals or continuously tracked.

Response: We have clarified this in the Methods.

Modified text:

“Comorbidities of psychiatric disorders and diabetes were identified through primary and secondary diagnoses from in- and outpatient care attended by specialists in the Swedish Patient Register using ICD codes (Supplementary Table 10). For diabetes, data on antidiabetic drug use from the Prescribed Drug Register (available from 2005) were also used.

In the analysis of cancer and CVD risk, we considered psychiatric disorders and diabetes as time-varying variables. Individuals with these diagnoses prior to January 1, 1985, or who never developed them by the end of the study period, were classified as having or not having the conditions throughout the follow-up, respectively. Individuals who developed these conditions during the follow-up period were classified as not having the conditions until diagnosis, and as having them thereafter.” (Methods section, page 13, lines 374-383)

“We adjusted for age and calendar period at follow-up (split yearly), psychiatric disorders, and diabetes as time-varying variables, as well as sex, birth year, country of birth, education, income, and employment status at the first blood sampling as time-fixed variables in the models.” (Methods section, page 15, lines 427-430)

5. Supplementary Tables: Supplementary Table 6 is critical to the main findings and could be moved into the main text or better summarized in the Results.

Response: We have now moved the original Supplementary Table 6 as Table 1 in the main manuscript.

Reviewer #2 (Remarks to the Author):

Summary

This manuscript investigates the impact of a cancer diagnosis on metabolic and inflammatory biomarkers, and their trajectories over time. It also examines the association between these trajectories and the risk of subsequent cardiovascular events (CVD). Using data from the Amoris cohort and linked Swedish national health registers, the authors present a well-conducted and clearly written study on an important and timely topic. The analyses are appropriate, the presentation is clear, and the findings may have clinical and public health relevance. I have some minor comments that may assist in strengthening the manuscript.

Recommendation

Minor revisions (if data for sensitivity analyses are not available).

Response: Thank you very much for the positive comments. We have addressed your comments point-by-point and revised the manuscript accordingly. The page and line numbers of the modified text refer to the clean version of the manuscript.

Comments

Definition of the Primary Outcome

o The manuscript refers to “cardiovascular disease” as the primary outcome. Based on the data sources used (e.g., inpatient register, cause of death registry), the outcome is likely restricted to major cardiovascular events requiring hospitalisation. If this is the case, please consider clarifying and/or modifying the terminology throughout the manuscript to avoid readers misinterpreting this as encompassing all CVD, including conditions that may be managed in primary care.

Response: We identified cardiovascular diseases from inpatient and hospital-based outpatient care records. Now we have clarified throughout the manuscript that our primary outcome is “hospital treated cardiovascular disease”.

Potential for Missed Outcomes

o Would the addition of medication prescription data results have improved case ascertainment? Is this possible for these analyses?

Response: We agree that incorporating medication prescription data could help capture some mild cases of cardiovascular disease. However, severe conditions, such as myocardial infarction, stroke, and heart failure—typically requiring hospital-based treatment—are unlikely to be misclassified when using data from the Swedish Patient Register, which was established in 1964 and has had nationwide coverage since 1987. In addition, medication prescription data from the Swedish Prescribed Drug Register have been available only since 2005, resulting in missing information for more than ten years following cancer diagnosis in our study.

We therefore chose to focus on relatively severe cases of CVD recorded in the Swedish Patient Register, and clarified this in the manuscript.

Modified text:

“The Patient Register has collected inpatient care data since 1964 (became nationwide in 1987) and outpatient care data since 2001⁵⁰” (Methods section, page 11, lines 327-328)

“Swedish Prescribed Drug Register have been available only since 2005, which limits the ability to use medication prescription data to identify milder CVD as the outcome of interest.” (Discussion section, page 11, lines 303-305)

“Therefore, we focused on relatively severe cases of CVD recorded in the Swedish Patient Register.” (Discussion section, page 11, lines 307-308)

Cancer Treatment–Specific Effects

o Different cancer treatments (e.g., cardiotoxic chemotherapies, radiotherapy that has damaged heart muscle) can influence CVD risk. If available, a sensitivity analysis stratified by treatment type would strengthen the study. If not feasible, please acknowledge this limitation explicitly.

Response: We agree that cancer treatment may influence CVD risk. However, data on cancer therapies were unfortunately not available in the Swedish Cancer Register. We have discussed the limitation in this manuscript.

Modified text:

“Cancer therapies are known to influence both CVD risk and metabolic biomarkers. For example, cardiotoxicity is a concern for many chemotherapies, especially anthracyclines and certain targeted agents (e.g., Trastuzumab and Tyrosine Kinase Inhibitors)²⁵. Chemotherapy, radiotherapy and hormone deprivation therapies have also been suggested to be associated with insulin resistance, dyslipidemia and hyperglycemia^{26–29}. Cytotoxic treatments can increase purine turnover through tumor cell lysis, even in the absence of overt tumor lysis syndrome³⁰. In addition, renal toxicity, nutritional changes, and systemic inflammation may alter metabolism or excretion of metabolic biomarkers^{31,32}. All of these might have contributed to the altered biomarker levels observed in cancer patients compared to non-cancer individuals in the present study, although the biomarker levels generally remained within the normal ranges. Nonetheless, as our trajectory analyses of biomarkers focused on long-term changes spanning across 12 years after a cancer diagnosis (Supplementary Figure 2), the potential influence of cancer therapies is limited.” (Discussion section, pages 7-8, lines 199-211)

“Last, we were unable to perform any stratification analysis by cancer treatment due to the limited availability of treatment data. Future research should expand these efforts to cancer patients receiving different therapies.” (Discussion section, page 11, lines 308-310)

Healthy Volunteer Bias

o As this is a secondary analysis of data from the Amoris cohort, it would be valuable to include discussion of the potential for healthy volunteer bias and/or other selection bias.

Response: About 90% of subjects between 20 and 64 years old were employed in AMORIS cohort (Walldius et al. 2017). There was a somewhat greater proportion of employed individuals in the AMORIS cohort compared with the general population, at the time of

recruitment, suggesting a potential 'healthy worker effect'. Although this does not affect the internal validity of studies based on the AMORIS cohort, it could lead to underestimations of prevalence and incidence rates in the general population. We have now added discussion in this regard in the manuscript.

Modified text:

"Second, this study was based on individuals who underwent occupational health check-ups or outpatient laboratory tests in Stockholm, Sweden. Although this does not affect the internal validity of studies derived from the AMORIS cohort, the generalizability of our findings to broader or more diverse populations (e.g., those of non-European ancestry or individuals in lower-resource contexts) may be limited. Further research in other populations and settings is warranted to validate our findings." (Discussion section, pages 10, lines 284-290)

Residual Confounding

o Given the observational design and constraints of the available data, residual confounding is likely. The discussion could more clearly convey that this is "more likely than not" rather than simply possible.

Response: We have discussed the issue of potential residual confounding in the manuscript.

Modified text:

"Third, although multiple covariates were adjusted for in the analyses, there may still be residual confounding from unknown or unmeasured factors, e.g., cancer therapies and sub-clinical cardiometabolic conditions not attended by specialist care, like atherosclerosis, dyslipidemia, dysglycemia, hyperuricemia, and their treatments. Therefore, caution is warranted when interpreting our findings and considering the clinical utility." (Discussion section, pages 10, lines 290-295)

Conclusion

Given the limitations of the study, the conclusion is somewhat strongly worded. Monitoring of blood biomarkers may be useful but further research is most certainly required before confirming this.

Response: we have modified our conclusion as suggested.

Modified text:

"In conclusion, cancer patients demonstrated different trajectories of glucose, fructosamine, TG, HDL, ApoA1, CRP, haptoglobin, and uric acid levels over time compared to cancer-free individuals. Cancer patients with specific trajectories of glucose, albumin, and uric acid had an increased risk of CVD. Further research on these blood biomarkers, through demonstrating underlying biological mechanisms and replication in independent cohorts, may contribute to the prevention of cardiovascular disease among cancer patients." (Discussion section, page 11, lines 311-316)

Minor Comments

Introduction (First Sentence)

o The statement on advances in early detection and treatment increasing survival could be rephrased for accuracy. For example: "Advances in early detection and cancer treatment have significantly increased survival rates for many cancer patients."

Response: We have changed the sentence as suggested.

Other

o Ensure consistent terminology for the primary outcome and biomarker names across the abstract, methods, results, and discussion.

Response: We have gone through the manuscript to modify this.

Reference

- Casco, Stephania, and Elena Soto-Vega. 2016. 'Development of Metabolic Syndrome Associated to Cancer Therapy: Review'. *Hormones & Cancer* 7(5–6):289. doi:10.1007/S12672-016-0274-1.
- Connor, Avonne E., Kala Visvanathan, Stephanie D. Boone, Nader Rifai, Kathy B. Baumgartner, and Richard N. Baumgartner. 2019. 'Fructosamine and Diabetes as Predictors of Mortality among Hispanic and Non-Hispanic White Breast Cancer Survivors'. *NPJ Breast Cancer* 5(1):3. doi:10.1038/S41523-018-0099-X.
- Faubert, Brandon, Ashley Solmonson, and Ralph J. DeBerardinis. 2020. 'Metabolic Reprogramming and Cancer Progression'. *Science* 368(6487). doi:10.1126/SCIENCE.AAW5473.
- Friedewald, W. T., R. I. Levy, and D. S. Fredrickson. 1972. 'Estimation of the Concentration of Low-Density Lipoprotein Cholesterol in Plasma, Without Use of the Preparative Ultracentrifuge'. *Clinical Chemistry* 18(6):499–502. doi:10.1093/CLINCHEM/18.6.499.
- Gudmundsdottir, Thorgerdur, Jeanette F. Winther, Sofie De Fine Licht, Trine G. Bonnesen, Peter H. Asdahl, Laufey Tryggvadottir, Harald Anderson, Finn Wesenberg, Nea Malila, Henrik Hasle, and Jørgen H. Olsen. 2015. 'Cardiovascular Disease in Adult Life after Childhood Cancer in Scandinavia: A Population-Based Cohort Study of 32,308 One-Year Survivors'. *International Journal of Cancer* 137(5):1176–86. doi:10.1002/IJC.29468.
- Hajar, Rachel. 2011. 'Statins: Past and Present'. *Heart Views : The Official Journal of the Gulf Heart Association* 12(3):121. doi:10.4103/1995-705X.95070.
- Hershey, Denise Soltow. 2017. 'Importance of Glycemic Control in Cancer Patients with Diabetes: Treatment through End of Life'. *Asia-Pacific Journal of Oncology Nursing* 4(4):313. doi:10.4103/APJON.APJON_40_17.
- Herzog, Katharina, Tomas Andersson, Valdemar Grill, Niklas Hammar, Håkan Malmström, Mats Talbäck, Göran Walldius, and Sofia Carlsson. 2022. 'Alterations in Biomarkers Related to Glycemia, Lipid Metabolism, and Inflammation up to 20 Years Before Diagnosis of Type 1 Diabetes in Adults: Findings From the AMORIS Cohort'. *Diabetes Care* 45(2):330–38. doi:10.2337/DC21-1238.
- Li, Qian, Guofu Zhang, Xiating Li, Shangzhi Xu, Huafang Wang, Jun Deng, Zhipeng Cheng, Fengjuan Fan, Shi Chen, Meng Yang, Liang V. Tang, and Yu Hu. 2025. 'Risk of Cardiovascular Disease among Cancer Survivors: Systematic Review and Meta-Analysis'. *EClinicalMedicine* 84:103274. doi:10.1016/j.eclinm.2025.103274.
- Malmström, Håkan, Göran Walldius, Valdemar Grill, Ingmar Jungner, Soffia Gudbjörnsdottir, and Niklas Hammar. 2014. 'Fructosamine Is a Useful Indicator of Hyperglycaemia and Glucose Control in Clinical and Epidemiological Studies – Cross-Sectional and Longitudinal Experience from the AMORIS Cohort'. *PLOS ONE* 9(10):e111463. doi:10.1371/JOURNAL.PONE.0111463.
- Psarakis, Helen M. 2006. 'Clinical Challenges in Caring for Patients With Diabetes and Cancer'. *Diabetes Spectrum* 19(3):157–62. doi:10.2337/DIASPECT.19.3.157.
- Ronit, A., D. M. Kirkegaard-Klitbo, T. L. Dohmann, J. Lundgren, C. A. Sabin, A. N. Phillips, B. G. Nordestgaard, and S. Afzal. 2020. 'Plasma Albumin and Incident Cardiovascular Disease: Results From the CGPS and an Updated Meta-Analysis'. *Arterioscler Thromb Vasc Biol* 40(2):473–82. doi:10.1161/atvbaha.119.313681.

- Rydén, Lars, and Jan Lindsten. 2021. 'The History of the Nobel Prize for the Discovery of Insulin'. *Diabetes Research and Clinical Practice* 175:108819. doi:10.1016/J.DIABRES.2021.108819.
- Seidu, S., S. K. Kunutsor, and K. Khunti. 2020. 'Serum Albumin, Cardiometabolic and Other Adverse Outcomes: Systematic Review and Meta-Analyses of 48 Published Observational Cohort Studies Involving 1,492,237 Participants'. *Scand Cardiovasc J* 54(5):280–93. doi:10.1080/14017431.2020.1762918.
- Tietze, Karen J. 2012. 'Chapter 5 - Review of Laboratory and Diagnostic Tests'. Pp. 86–122 in *Clinical Skills for Pharmacists (Third Edition)*, edited by K. J. Tietze. Saint Louis: Mosby.
- Toyoshima, Marcos Tadashi Kakitani, Priscilla Cukier, Aline Santos Damascena, Rafael Loch Batista, Fernanda de Azevedo Correa, Eduardo Zanatta Kawahara, Carlos André Minanni, Ana O. Hoff, and Marcia Nery. 2023. 'Fructosamine and Glycated Hemoglobin as Biomarkers of Glycemic Control in People with Type 2 Diabetes Mellitus and Cancer (GlicoOnco Study)'. *Clinics* 78:100240. doi:10.1016/J.CLINSP.2023.100240.
- Treciokiene, Indre, Kamile Daukintyte, Paul Hjemdahl, and Björn Wettermark. 2024. 'Trends in Statin Utilization and Ischemic Heart Disease Mortality in Lithuania and Sweden, 2000–2020'. *Upsala Journal of Medical Sciences* 129:10.48101/ujms.v129.10412. doi:10.48101/UJMS.V129.10412.
- Trehan, Shubam, Gurjot Singh, Adarshpreet Singh, Gaurav Bector, Aayush Jain, Priya Antil, Fnu Kalpana, Amna Farooq, and Harmandeep Singh. 2024. 'Chemotherapy and Metabolic Syndrome: A Comprehensive Review of Molecular Pathways and Clinical Outcomes'. *Cureus* 16(8):e66354. doi:10.7759/CUREUS.66354.
- Walldius, Göran, Ingmar Jungner, Ingar Holme, Are H. Aastveit, Werner Kolar, and Eugen Steiner. 2001. 'High Apolipoprotein B, Low Apolipoprotein A-I, and Improvement in the Prediction of Fatal Myocardial Infarction (AMORIS Study): A Prospective Study'. *The Lancet* 358(9298):2026–33. doi:10.1016/S0140-6736(01)07098-2.
- Winther, Jeanette F., Smita Bhatia, Luise Cedervist, Thorgerdur Gudmundsdottir, Laura Madanat-Harjuoja, Laufey Tryggvadottir, Finn Wesenberg, Henrik Hasle, and Anna Sällfors Holmqvist. 2018. 'Risk of Cardiovascular Disease among Nordic Childhood Cancer Survivors with Diabetes Mellitus: A Report from Adult Life after Childhood Cancer in Scandinavia'. *Cancer* 124(22):4393–4400. doi:10.1002/CNCR.31696.
- Wulaningsih, Wahyu, Lars Holmberg, Hans Garmo, Björn Zethelius, Annette Wigertz, Paul Carroll, Mats Lambe, Niklas Hammar, Göran Walldius, Ingmar Jungner, and Mieke Van Hemelrijck. 2013. 'Serum Glucose and Fructosamine in Relation to Risk of Cancer'. *PLOS ONE* 8(1):e54944. doi:10.1371/JOURNAL.PONE.0054944.
- Xu, Liuyue, Qian Dong, Yaoying Long, Xiaoqiong Tang, Nan Zhang, and Kai Lu. 2020. 'Dynamic Changes of Blood Lipids in Breast Cancer Patients After (Neo)Adjuvant Chemotherapy: A Retrospective Observational Study'. *International Journal of General Medicine* 13:817. doi:10.2147/IJGM.S273056.

REVIEWERS' COMMENTS

Reviewer #1 (Remarks to the Author):

The authors have adequately addressed the major concerns raised in the previous round of peer review. Specifically:

- They now explicitly acknowledge the lack of data on cancer treatment modalities and recognize this as a major limitation that could influence biomarker trajectories and cardiovascular risk.
- They have clarified the short-term validity and use of fructosamine as a glycemic biomarker, and added discussion regarding its clinical relevance in oncology.
- The transient increase in leukocytes and LDL/HDL ratio following cancer diagnosis is now correctly attributed to possible treatment-related effects.
- The absence of statin use and other cardiometabolic medications has been added as a limitation.
- The authors also clarified that their findings are primarily prognostic in nature, not necessarily causal.

While the manuscript could further benefit from an explicit comparison to prior AMORIS-based studies (e.g., Wulaningsih et al., 2013; Malmström et al., 2014) and from a more detailed discussion on the clinical applicability of latent trajectory modeling, these are secondary issues that do not detract significantly from the overall strength of the study.

The manuscript is methodologically sound, clinically relevant, and provides valuable insights into longitudinal biomarker patterns and cardiovascular risk in cancer survivors.

I believe the manuscript is now suitable for acceptance in its current form or with minor editorial adjustments.

Response: Thank you for the positive comments. We have further highlighted the comparison of the current study to prior AMORIS-based studies from the perspective of latent trajectory modeling. Modified text:

" Fructosamine has been suggested to be a glycemic marker in the AMORIS cohort ³⁶. Mean of 4 repeated measurements of fructosamine after enrollment to AMORIS, before or after cancer diagnosis, was associated with a reduced cancer risk ³⁷ , while a single measurement of fructosamine around 5 months

post-diagnosis was linked to an increased risk of death among breast cancer patients³⁸. These conflicting findings may suggest that elevated fructosamine levels reflect different underlying metabolic or nutritional states, although both studies have relied on single values of fructosamine. In contrast, we investigated longitudinal patterns of this biomarker following cancer diagnosis and found that, compared to cancer-free individuals, cancer patients showed consistently high levels during the 12 years after cancer diagnosis.” (lines 243-251 in the clean version of the manuscript)

”Taking advantage of the repeated measurements of biomarkers, the current study expands the knowledge for AMORIS cohort by using latent trajectory modeling.” (lines 255-256)

Reviewer #3 (Remarks to the Author):

Thank you for the opportunity to review this manuscript. The authors have done a very good job of responding to the thoughtful and extensive comments raised by the reviewers so far. I had a few suggestions to improve the clarity of the work.

The results show that the risk increase of CVD was highest if the cancer was diagnosed before age 18. It would be useful to explicitly state that this was based on 60 cancers in that age group.

Response: We have clarified this in the manuscript. Modified text:

“Additionally, we evaluated the association by age at cancer diagnosis, for subtypes of CVD and subtypes of cancer (organ systems or sites) separately.” (lines 438-440)

“The risk increase of CVD was highest if the cancer was diagnosed before age 18 [IRR (95% CI): 2.54 (1.97-3.28)], based on all studied cancer types.” (lines 96-97)

The part in the discussion about age could be written more clearly. Their finding does seem to fit with previous work (childhood cancer patients diagnosed before 18 years have been shown to experience approximately two- to three-fold greater CVD risk) but this is not clearly compared. The statement "It would be clinically meaningful to examine whether post-cancer biomarker trajectories differ by age at diagnosis and whether their implications vary across age groups. However, we were unable to do so due to lack of data on childhood cancer patients since the included participants were

diagnosed with cancer at a mean age of 60.7" is not very clear. I think what the authors mean is that the numbers were too small in some age groups to support detailed analysis by age at diagnosis.

Response: We have further clarified the discussion about age. As suggested, we have now highlighted that the current study and the previous literature are consistent, and we were unable to perform analysis on trajectories due to small numbers. Modified text:

"In line with prior findings^{23,24}, the present study also found that childhood cancer patients (e.g., diagnosed before 18 years) showed to experience a substantially higher risk of CVD after cancer diagnosis—approximately two- to three-fold greater than that of cancer-free individuals. Given that childhood cancer patients have a relatively longer life expectancy than those patients diagnosed in adulthood, it would be clinically meaningful to examine whether post-cancer biomarker trajectories differ by age at diagnosis and whether their implications vary across age groups. However, we were unable to do so due to small numbers of childhood cancer patients included in this study." (lines 193-200)

I find the inclusion of both organ system and site of cancer a little confusing, as the results switch from one to the other. It would be good to explicitly state that Analysis 1 is broken down by site, but due to numbers Analysis 2 and 3 could only be broken down by system (if that was the case). Or perhaps just stick with organ system throughout?

Response: We have further clarified this in the manuscript as suggested.

Modified text:

"Due to small numbers, cancer types were broken down by system in Analysis 2 and 3 instead of both system and cancer site as in Analysis 1." (lines 468-470)

More of an editorial comment, but there are some very long paragraphs in the discussion and I would recommend breaking some of these up to aid readability.

Response: We have broken up the Discussion section as suggested.